# The effect of climate type on timescales of drought propagation in an ensemble of global hydrological models

Anouk I. Gevaert[1], Ted I. E. Veldkamp[2,3], Philip J. Ward[2]

[1] Faculty of Science, Vrije Universiteit Amsterdam, the Netherlands
[2] Institute for Environmental Studies (IVM), Vrije Universiteit Amsterdam, the Netherlands
[3] International Institute for Applied Systems Analysis (IIASA), Laxenburg, Austria

*Correspondence to*: Ted I. E. Veldkamp (ted.veldkamp@vu.nl)

**Abstract.** Drought is a natural hazard that occurs at many temporal and spatial scales and has severe environmental and socio-economic impacts across the globe. The impacts of drought change as drought evolves from precipitation deficits to
deficits in soil moisture or streamflow. Here, we quantified the time taken for drought to propagate from meteorological drought to soil moisture drought, and from meteorological drought to hydrological drought. We did this by cross-correlating the Standardized Precipitation Index (SPI) against standardized indices of soil moisture, runoff, and streamflow from an ensemble of global hydrological models forced by a consistent meteorological dataset. Drought propagation is strongly related to climate types, occurring at sub-seasonal timescales in tropical climates and at up to multi-annual timescales in
continental and arid climates. Winter droughts are usually related to longer SPI accumulation periods than summer droughts, especially in continental and tropical savanna climates. The difference between the seasons is likely due to winter snow cover in the former and distinct wet and dry seasons in the latter. Model structure appears to play an important role in model variability, as drought propagation to soil moisture drought is slower in land surface models than in global hydrological models, but propagation to hydrological drought is faster in land surface models than in global hydrological models. The
propagation time from SPI to hydrological drought in the models was evaluated against observed data at 127 in-situ streamflow stations. On average, errors between observed and modeled drought propagation timescales are small and the model ensemble mean is preferred over the use of a single model. Nevertheless, there is ample opportunity for improvement as substantial differences in drought propagation are found at 10 % of the study sites. A better understanding and representation of drought propagation in models may help improve seasonal drought forecasting as well as constrain drought
variability under future climate scenarios.

## 1 Introduction

Drought is a complex global phenomenon with severe environmental (Bond et al., 2005; Lewis and Sjöstrom, 2010; Reichstein et al., 2013; Turco et al., 2017; Vicente-Serrano et al., 2013) and socio-economic (Horridge et al., 2005; Stanke et al., 2013; Wegren, 2011) impacts. Data from global and regional models are commonly used to study droughts (e.g.

Andreadis and Lettenmaier, 2006; Sheffield et al., 2004; Sheffield and Wood, 2008a) and water scarcity (Kummu et al., 2016; Veldkamp et al., 2015, 2016, 2017; Wada et al., 2011), and how these will change in the future (Dai, 2013; Huang et al., 2017; Sheffield and Wood, 2008b; Trenberth et al., 2014). However, the evolution of drought from meteorological anomalies to deficits in soil moisture and streamflow in these models is still poorly understood (Van Lanen et al., 2013; Van Loon, 2015), despite the fact that drought impacts are more closely related to these latter components of the hydrological cycle (Van Loon, 2015).

Several types of drought can be distinguished (i.e. Van Loon, 2015; Mishra and Singh, 2010). The first stage or type of drought is called meteorological drought, and is caused by precipitation deficits that may or may not be combined with above-normal potential evaporation rates. If the precipitation deficits and/or increased evaporation rates are sustained for a sufficient period, they can result in lower than average soil moisture availability, which may result in a soil moisture drought. In the same way, meteorological drought can propagate into lower streamflow and groundwater levels, which are both forms of hydrological drought. Another type of drought, socio-economic drought, is not based on a hydrological variable alone, but occurs when water availability is lower than water demand. As drought propagates from meteorological drought to soil moisture or hydrological drought, the characteristics of droughts change. As drought moves through components of the hydrological cycle, the onset of droughts tends to be lagged, there tend to be fewer but longer drought events, and the droughts are attenuated (Van Lanen et al., 2013; Van Loon et al., 2012; Peters et al., 2003). The degree to which these drought propagation characteristics are observed depends on climate and catchment properties (Van Lanen et al., 2013; Van Loon and Laaha, 2015).

Several drought propagation studies have been carried out based on observational data at catchment to national scales, mainly focusing on meteorological and streamflow drought (e.g. Barker et al., 2016; Haslinger et al., 2014; Van Loon and Laaha, 2015; Lorenzo-Lacruz et al., 2013). However, the geographical extent covered by these studies is limited, mainly due to the lack of observational data with sufficiently long timescales needed to identify drought in many regions. Therefore, data from regional or global models are generally used to study drought at larger spatial scales. In some cases, studies have used data from a single large-scale model (e.g. Van Lanen et al., 2013; Lehner et al., 2006; Lloyd-Hughes et al., 2013; Sheffield et al., 2004). However, there can be considerable differences between hydrological outputs of different large-scale models (Burke and Brown, 2008; Gudmundsson et al., 2012b, 2012a; Haddeland et al., 2011; Prudhomme et al., 2014; Veldkamp et al., 2018), suggesting that multi-model approaches (e.g. Gudmundsson et al., 2012a; Stahl et al., 2012; Tallaksen and Stahl, 2014; Wang et al., 2009) are more appropriate. This is supported by the fact that studies have found that a model ensemble performs better than individual models (Gudmundsson et al., 2012a; Stahl et al., 2012). However, most regional or global drought studies have focused on drought frequency and severity (van Huijgevoort et al., 2013; Prudhomme et al., 2011, 2014) rather than on drought propagation, and those that have studied drought propagation have been limited to a single model (Van Lanen et al., 2013) or to a small selection of contrasting catchments (Van Loon et al., 2012).

The aim of this study is therefore to investigate drought propagation times in an ensemble of global hydrological models from meteorological drought to soil moisture drought, and from meteorological drought to hydrological drought. We focus on the effect of climate on drought propagation in particular, and distinguish between summer and winter droughts. In Sect. 2, we describe the identification of drought, the method we use to quantify drought propagation timescales, and the validation analysis. The global models and the observational data are presented in Sect. 3. In Sect. 4, we first describe and discuss the timescales of drought propagation and their relationship with climate types in the multi-model ensemble mean. Second, we briefly describe the model variability. Third, we perform a validation exercise to gauge whether propagation from meteorological to streamflow drought in the models resembles observational drought propagation times. We wrap up with a brief summary and conclusion in Sect. 5.

## 2 Methods

The timescales of drought propagation were determined by relating standardized indices of meteorological drought to indices of soil moisture and hydrological drought. These indices were derived from an ensemble of seven global models that were forced with a consistent meteorological dataset in the eartH2Observe project (Schellekens et al., 2016). For more details on the model and forcing datasets, see Sect. 3. First, we calculated standardized indices of precipitation, soil moisture, runoff, and streamflow as a measure of drought. Second, we determined which timescales of meteorological drought were most strongly correlated to the other drought types. Finally, we evaluated the modeled drought propagation times against drought propagation times derived from observational data.

### 2.1 Drought indices

Meteorological drought was quantified by the widely used Standardized Precipitation Index (SPI; Mckee et al. 1993). The SPI fits a pre-defined probability distribution to the frequency distribution of precipitation. Generally, the precipitation series have a monthly resolution, in which case the fitting is done for each month separately. Examples of possible distribution functions are the gamma, Pearson Type III, or log-normal (Guttman, 1999; Lloyd-Hughes and Saunders, 2002; Mckee et al., 1993) functions, though a non-parametric approach has also been developed (Hao and AghaKouchak, 2014). In this study, we use the gamma distribution as the pre-defined probability distribution, as this is commonly considered to be the most suitable (Lloyd-Hughes and Saunders, 2002; Mckee et al., 1993; Stagge et al., 2015). Regions in which months with zero precipitation are common are problematic for the SPI (Wu et al., 2007), so the cumulative distribution function is corrected with the probability of zero precipitation (Naresh Kumar et al., 2009). The fitted probability distribution is then transformed to the normal distribution, resulting in negative (positive) values for dry (wet) conditions. Note that the SPI relates the precipitation amount in a certain month to average conditions at that particular location. In drier climates or seasons, water-limited conditions may be the norm, while in wetter climates or seasons, water stress may not be relevant until severe drought thresholds are reached.

Advantages and disadvantages of the SPI are described in Hayes et al. (1999). A first advantage is that the SPI is relatively simple, requiring only precipitation data as an input. Secondly, the index is flexible. It can be used to compare different timescales of drought by aggregating the input precipitation time series over a number of months, known as the accumulation period (typically 1, 3, 6, 12, or 24 months). Thirdly, the standardization facilitates comparisons of extreme conditions in different locations, but also over different timescales. A disadvantage is that the precipitation data used to calculate the SPI may not be representative of surface conditions. In addition, the calculation of the index requires long time series (>30 years) of data (Guttman, 1999; Mckee et al., 1993; Zargar et al., 2011).

The standardization approach of the SPI is not only easily applied to different timescales, but can also be applied to other (hydrological) variables such as soil moisture or streamflow. In this way, we use the same methodology to identify soil moisture and hydrological drought by defining the Standardized Soil Moisture Index (SSMI) (Hao and AghaKouchak, 2013), Standardized Runoff Index (SRI) (Shukla and Wood, 2008), and Standardized Streamflow Index (SSFI).

Soil moisture and runoff are controlled by pixel-scale precipitation and hydrological processes. Streamflow, on the other hand, is affected by upstream pixels. Therefore, we computed a catchment-aggregated SPI to quantify drought propagation from meteorological to streamflow droughts. The input for the aggregated SPI is the total precipitation falling within each pixel and its upstream area, based on a 0.5° routing network (see Sect. 3). We did not include a travel time factor in this calculation, thus assuming that precipitation falling in the upper parts of each catchment will impact streamflow at the outlet within the same month. In the rest of the paper, meteorological drought is based on the pixel-based SPI when referring to soil moisture or runoff drought, and to the catchment-based SPI when referring to streamflow drought.

In this study, we quantified drought propagation from meteorological drought to soil moisture and hydrological droughts. Therefore, we calculated the SPI using accumulation periods of 1, 2, 3, 6, 9, 12, 24 and 36 months. These accumulation periods span sub-seasonal, (multi-)seasonal, and (multi-)annual timescales. For the other standardized indices (SIs), we used the 1-month accumulation period to identify short-term drought conditions.

## 2.2 Timescales of drought propagation

Drought propagation from meteorological to soil moisture or hydrological drought was based on correlations between the SPI and target SI (SSMI, SRI or SSFI). The SI time series were prepared by applying two criteria to the target SI. First, we distinguished between drought conditions in summer and winter seasons. The summer (winter) season was defined as June, July, and August above (below) the equator, and December, January, and February below (above) the equator. Second, we focused on months corresponding to dry conditions, here defined as SI $\leq 0$. This threshold includes near-normal and mildly dry conditions, but leaves a larger sample size than when we limit the analysis to moderate or severe drought events. Theoretically, moderate drought events (SI $\leq -1$) have an occurrence probability of about 16 % per year, which would correspond to about 5 drought months in a 30-year study period, for a total of 15 drought months considering a three-month season. Severe drought events (SI $\leq -1.5$) would result in six drought months during a 30-year period of three months based on the same reasoning, compared to 45 events using SI $\leq 0$ as a threshold.

After preparing the SI time series, we cross-correlated each of the SPI time series with the SSMI, SRI, and SSFI, without considering lag between the time series. The drought propagation timescale was defined as the SPI accumulation period that is most closely related to the target drought index, which we call SPI-n. In this analysis we determine which SPI accumulation period best represents drought propagation timescales overall during the study period. However, drought

propagation may occur at different time scales in the same location and season, and any specific meteorological drought event may propagate to other droughts more quickly, or more slowly, than suggested by SPI-n. Pixels where the final correlations between SPI-n and SSMI, SRI, or SSFI are not statistically significant ($p = 0.05$) were masked from the results. Autocorrelation is a potential issue when correlating time series, as it reduces the degrees of freedom compared to a standard significance test. In this study, the effective degrees of freedom are based on the modified Chelton method (Pyper and

Peterman, 1998) as in Barker et al. (2016).

The strength of the relationships between the Köppen-Geiger climate type classification (Kottek et al., 2006) shown in Fig. 1, as well as certain model characteristics on SPI-n, were quantified using a variety of tests. We used the rank of SPI-n rather than the duration of the accumulation period in months in the calculations. This means we assumed that the difference between accumulation periods of 12 and 24 months (both in the order of annual timescales, differing by one rank SPI-n) to

be equivalent to the difference between 1 and 2 months (both in the order of sub-seasonal timescales, differing by one rank SPI-n), but very unlike the difference between 1 and 12 months (sub-seasonal versus annual timescales, differing by 5 rank SPI-n). The choice of statistical tests is not straightforward. Rank SPI-n are essentially ordinal variables, which are tested using metrics such as Chi-squared. However, Chi-squared and comparable tests treat ordinal variables as categorical variables. In our case this means that the relationships between the used accumulation periods are not taken into account.

Since the chosen accumulation periods are nearly equidistant in log space, we chose to apply statistical tests developed for interval data in addition to Chi-squared. Statistical significance was based on ANOVA tests, with the Tukey's honestly significant difference test as a post-hoc test to compare groups in a pairwise fashion. Tukey's test corrects for family-wise errors, or the fact that the chance of type 1 errors increases when comparing multiple groups. When only two groups are possible we used paired t-tests.

With large quantities of data, even very small differences between group means can be statistically significant. Measures of effect size are more useful to investigate the magnitude of the differences between groups, which may be more relevant for interpretation and policy. We use Cohen's d to quantify the effect size between two groups (Cohen, 1988). This metric is defined as:

$$d = \frac{\mu_1 - \mu_2}{\sigma_{pooled}} \tag{1}$$

where

$$\sigma_{pooled} = \sqrt{\frac{(n_1-1)\sigma_1^2 + (n_2-1)\sigma_2^2}{n_1+n_2-2}} \tag{2}$$

and where μ represents the group mean, here mean rank SPI-n, σ the standard deviation of rank SPI-n, $n$ the number of observations, and subscripts indicate the group, here climate types. The outcome of the metric is thus the difference in group means relative to the standard deviation of the groups. A result of 1 means that the group means differ by one standard deviation and thus that those groups overlap by about 62 %. Previous studies tend to use fixed thresholds to interpret small (0.2), medium (0.5) and large (0.8) effect sizes, though these thresholds are considered to be rather arbitrary (Cohen, 1988; Lenth, 2001).

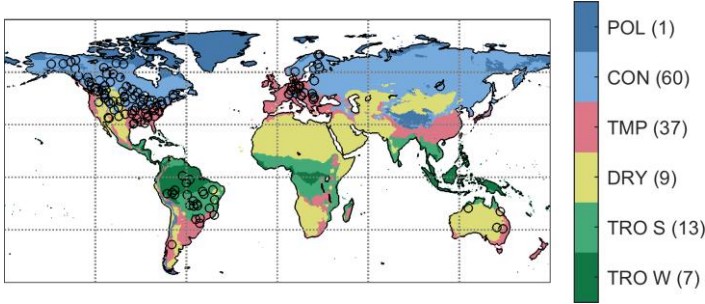

**Figure 1: Map of the Köppen-Geiger climate classification and the GRDC stations used in this study. The number of GRDC stations within each climate type is included in the legend in brackets. TRO W = tropical wet, TRO S = tropical savanna, DRY = dry, TMP = temperate, CON = continental, POL = polar.**

## 2.3 Validation of drought propagation

Timescales of drought propagation from meteorological to streamflow drought in global hydrological models are validated against observational data. Sites with observational streamflow data were matched to model pixels by selecting the model pixel containing the in-situ site. Since there may be discrepancies between the model and actual river routing schemes, we extracted the model streamflow data from the selected pixel as well as from the eight surrounding pixels. The in-situ site was then assigned to the model pixel with the lowest root mean square error (RMSE) between observed and modeled monthly streamflow. Once the in-situ sites were matched to model pixels, the SPI-n were calculated as described in Sect. 2.2. Since the observational time series may have gaps within the study period, model results were also recalculated at the in-situ sites using only months for which observational data were available.

Model discharge time series from the models used in this study (see Sect. 3) have been evaluated in previous studies (Beck et al., 2017; Schellekens et al., 2016), so we limit the evaluation to drought propagation characteristics, or SPI-n. The evaluation of SPI-n was based on the rank SPI-n rather than the length of the accumulation period in months, for the same reason for which rank SPI-n were used in the statistical tests (Sect. 2.2). The performance metrics used in the evaluation were mean absolute error (MAE) and Spearman correlation coefficient.

## 3 Data

We assessed drought propagation in seven global models from the eartH2Observe project ([www.earth2observe.eu](www.earth2observe.eu)), as well as in the model ensemble mean. Three of the models are land surface models (LSMs): HTESSEL-CaMa (Balsamo et al., 2009; Yamazaki et al., 2011), ORCHIDEE (D'Orgeval et al., 2008; Krinner et al., 2005; Ngo-Duc et al., 2007), and the
SURFEX-TRIP modeling platform (Decharme et al., 2010, 2013). The other four models are Global Hydrological Models (GHMs): LISFLOOD (Van Der Knijff et al., 2010), PCR-GLOBWB (van Beek et al., 2011; Wada et al., 2014), W3RA (van Dijk, 2010; Van Dijk et al., 2014), and WaterGAP3 (Döll et al., 2009; Flörke et al., 2013). Other models in the eartH2Observe project were excluded because they did not provide all of the variables required for this study. The models are run with a consistent 0.5° meteorological forcing dataset, the WATCH Forcing Data methodology applied to ERA-
Interim reanalysis (WFDEI) data (Weedon et al., 2014). However, static fields such as land cover and soil physical properties were not prescribed, as these tend to be closely linked to the modeling system. Important characteristics of the models such as runoff mechanisms and representation of reservoirs or water use are presented in Table 1. For more information about the model datasets and project design, see Schellekens et al. (2016). Since we cannot disentangle the effects of the differences in model structures and parameterizations in the current experimental design, we focus on the
results of the model ensemble mean rather than the individual models. The ensemble mean provides insight into the model consensus on drought propagation time scales and how these vary by climate. Nevertheless, we present the individual model results in the Supplementary Information.

In this study, we used the monthly precipitation, root-zone soil moisture, runoff, and streamflow datasets to calculate the SIs. We do not study groundwater droughts because HTESSEL-CaMa does not simulate this store, and two of the other models
have not made the data available. In addition, groundwater is defined differently between models, complicating comparisons of this store. The common forcing dataset means that the SPI time series are identical for all models, while the SSMI, SRI, and SSFI are model-specific. The model ensemble mean was calculated as the average of the SIs. This method deviates from the standard approach in which the ensemble mean is the average of the original model time series. We have chosen to average SI time series because root-zone soil moisture storage and its variability vary considerably between models in
certain regions. In this way, models with high soil moisture (variability) have a much larger influence on the mean time series than models with low soil moisture (variability). In these situations averaging SI time series better captures the overall model response than averaging original model time series (see Fig. S1 for an example). Though averaging SI time series will result in a narrower range of values for the ensemble mean than for the individual models, the outcome of the SPI-n analysis is not greatly affected because it is based on correlations. A consistent model dataset for the eartH2Observe project is
available from 1980–2012, though we use the years 1983–2012 to avoid data gaps in the first years of the SI time series caused by the 36-month accumulation period for the SPI. In addition to the datasets used to calculate the SIs, we used other model variables to relate these to the drought propagation results. These are the runoff coefficient, or the ratio of runoff to precipitation, and the ratio of surface runoff to total runoff.

**Table 1: Overview of models and relevant characteristics**

| Model | Model type | Evaporation | Snow | Soil layers | Runoff | Reservoirs | Water use |
|---|---|---|---|---|---|---|---|
| HTESSEL-CaMa | LSM | Penman-Monteith | Energy balance | 4 | Saturation excess | No | No |
| LISFLOOD | GHM | Penman-Monteith | Degree-day | 2 | Saturation and infiltration excess | Yes | Yes |
| ORCHIDEE | LSM | Bulk method (Barella-Ortiz et al., 2013) | Energy balance | 11 | Green-Ampt infiltration | No | Irrigation only |
| PCR-GLOBWB | GHM | Hamon | Temperature based | 2 | Saturation excess | Yes (lakes only) | No |
| SURFEX-TRIP | LSM | Penman-Monteith | Energy and mass balance | 14 | Saturation and infiltration excess | No | No |
| W3RA | GHM | Penman-Monteith | Degree-day | 3 | Saturation and infiltration excess | No | No |
| WaterGAP3 | GHM | Priestley-Taylor | Degree-day | 1 | Beta function | Yes | Yes |

The validation of modeled drought propagation was based on observed precipitation and streamflow time series. We used gauge-based precipitation data from the Global Precipitation Climatology Centre's (GPCC) Full Data Reanalysis product version 7 (Schneider et al., 2015). The data are available as monthly precipitation totals at 0.5° resolution for the entire modeled period. Note that reanalysis precipitation data, such as used for the model forcing in this study, and the GPCC dataset are not truly independent. However, the dependence of reanalysis precipitation on both gauge and satellite observations inhibits the selection of a completely independent dataset with global coverage that also has a record long enough for drought studies. Monthly streamflow data were obtained from the Global Runoff Data Centre (GRDC; Koblenz, Germany; http://grdc.bafg.de) database. We used only sites with a catchment area larger than 9000 km$^2$, where the model upstream catchment area is within 25 % of the GRDC upstream catchment area, and which have at least 15 years of data. In addition, we ensured that the sites were independent by searching for the most upstream stations that fit the previous requirements. All stations located downstream of these stations were excluded from the analysis. Finally, we only report on sites where the correlation between SPI-n and the SSFI was statistically significant ($p < 0.05$). These criteria resulted in 127 sites (see Fig 1), with an average data availability of 29 years.

## 4 Results and discussion

Here, we characterize and discuss the timescales of drought propagation from meteorological to soil moisture and hydrological drought at global scale. First, we describe drought propagation in the model ensemble mean and its relationship to climate. This gives us an idea of the model consensus. Second, we assess the variability between models and identify factors that may explain these differences. Finally, we compare the results of the model ensemble mean as well as the individual models to observational data.

## 4.1 Model ensemble mean

Drought propagation based on SPI-n for the model ensemble mean varies considerably in space, and all timescales from 1 to 36 months are represented in the results (Fig. 2). Summer soil moisture droughts (based on SSMI) are best represented by SPI-n of one or two months in wet regions such as the Amazon, but by much longer SPI-n in dry climates and some boreal regions. Results are more mixed when focusing on runoff droughts (based on SRI). Runoff droughts are most linked to precipitation deficits in the same month in dry climates such as the Sahel, southern Africa, and central Australia. Runoff droughts in other dry regions such as the Middle East, northern Africa, and the western USA, however, are related to much longer precipitation deficits up to several years. The patterns of drought propagation timescales from meteorological to streamflow drought are similar to the patterns of SPI-n for runoff droughts, though SPI-n tends to be slightly longer. Longer SPI-n for streamflow compared to runoff can be expected as streamflow is simply routed runoff. In general, SPI-n are also longer, and drought propagation slower, for winter droughts than for summer droughts (Fig. 2). In this study, we focus on SPI-n rather than the strength of the relationship between SPI and other SIs. However, the correlations behind SPI-n are generally high, with median values ranging from 0.67 to 0.74, depending on climate and season (Fig. S2). The strength of the correlation is highest in tropical climates (medians around 0.8) and lowest in polar climates (medians around 0.6).

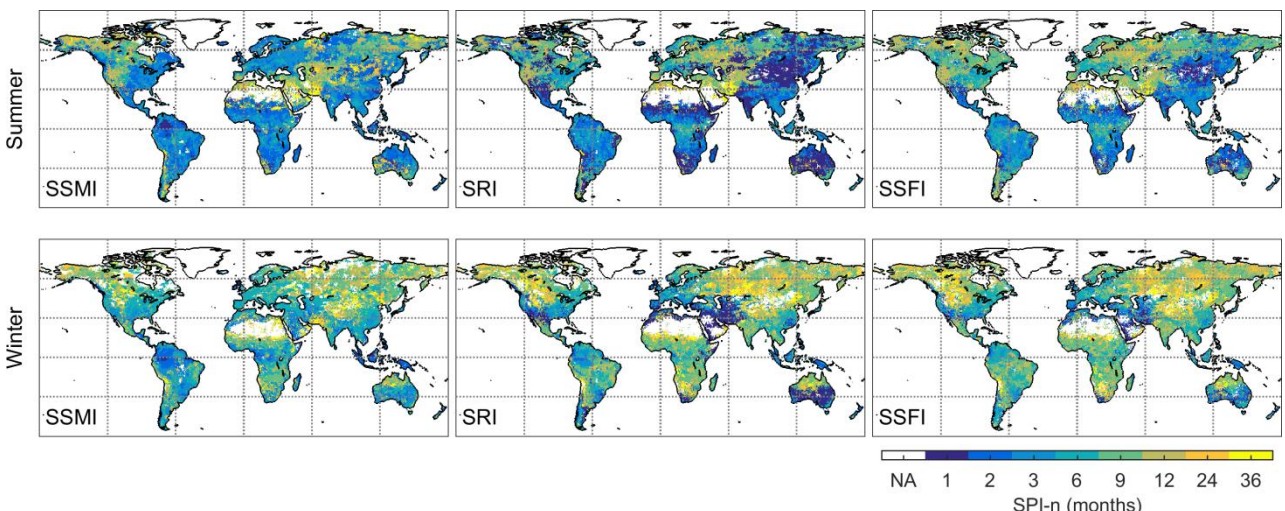

**Figure 2: The SPI accumulation period (SPI-n) resulting in the highest correlations with model ensemble mean SSMI, SRI, and SSFI, for summer and winter droughts. Pixels where those correlations are not statistically significant (p < 0.05) are masked.**

5    The relationship between drought propagation timescales and climate types is further examined using the Köppen-Geiger classification (Kottek et al., 2006). We use six climate classes that reflect the five major climate types, with an additional distinction between tropical wet (i.e. tropical rainforest or monsoonal climates) and tropical savanna climates (Fig. 1). The results in Fig. 3 confirm that climate type plays an important role in the timescale of drought propagation. As in Fig. 2, droughts in tropical climates tend to respond to short periods of accumulated precipitation deficits, while continental and

10   polar climates respond to longer periods of accumulated precipitation deficits. Overall, the variability within the tropical climate groups (both wet and savanna) is relatively low compared to dry climates, which are represented by the entire range of SPI accumulation periods studied. Despite the large variability in drought propagation timescales in dry climates, further distinctions between desert/savanna or hot/cold climates within the dry climate class did not have added value.

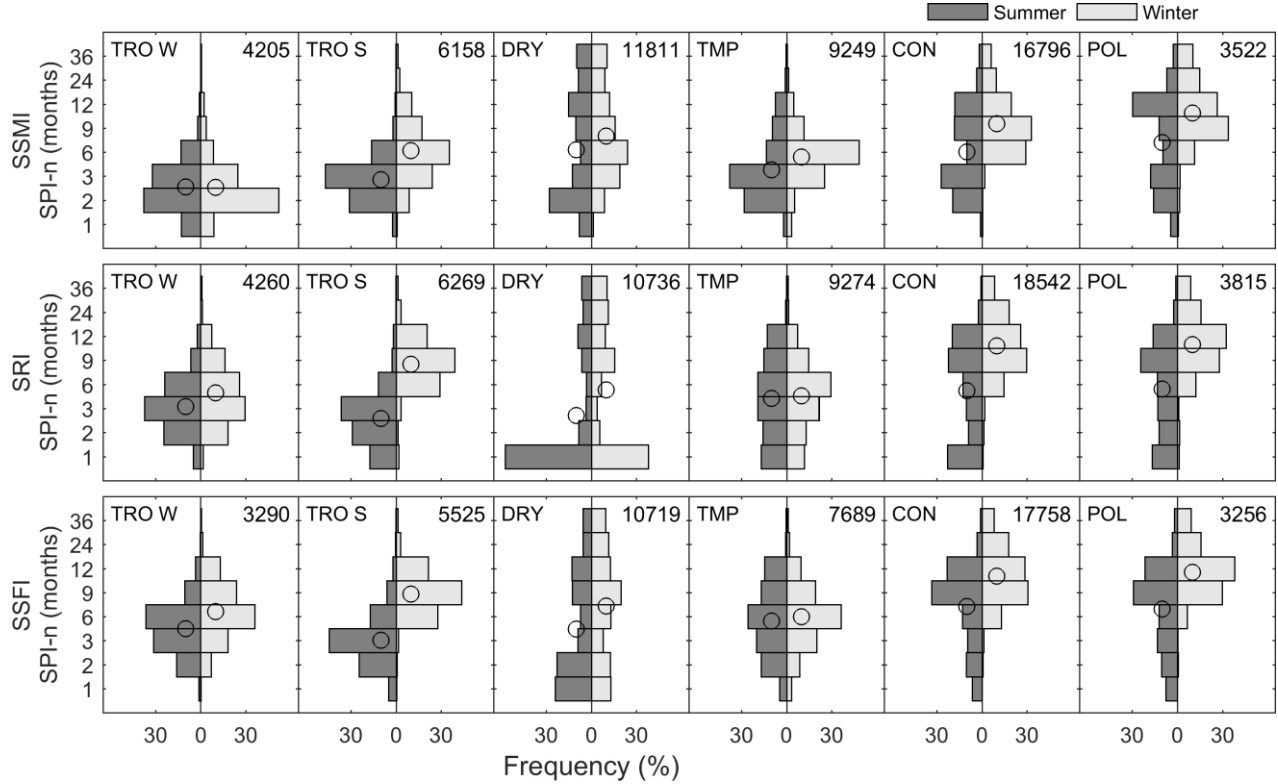

**Figure 3: Histograms of the SPI-n in months by climate type, and for summer and winter droughts in SSMI, SRI, and SSFI. Circles represent the mean rank SPI-n per climate type and season and numbers in the upper right indicate the number of summer and winter data pairs per climate and drought type. Abbreviations of climate types are the same as in Fig. 1.**

Winter droughts are related to longer SPI-n than summer droughts in tropical savanna, continental and polar climates. Soil moisture droughts in tropical wet climates, and hydrological droughts in temperate climates, on the other hand, have similar propagation characteristics in summer and winter. The seasonality of SPI-n in tropical savanna climates may be related to the distinct wet (summer) and dry (winter) seasons in these regions: SPI-n are more similar to tropical wet climates in summer

10 and to dry climates in winter. In continental climates, longer SPI-n may be due to snow cover in the winter. Precipitation in the form of snow during fall and winter will not replenish soil moisture or runoff until temperatures rise sufficiently to melt the snow. In this way, drought conditions in winter may be more related to precipitation deficits in the previous summer. Indeed, soil moisture droughts may be more related to the SPI accumulated over the three-month period before snowfall than to the longer period starting with those three months and extending until the defined winter season. Note that we used a

15 rather simple definition of summer and winter seasons by using the equator as divider. However, since the climate along the equator is almost exclusively tropical wet, which does not show large seasonality, we expect that this does not significantly impact the results.

In terms of statistical significance, ANOVA and Chi-squared tests show that the differences in mean rank SPI-n between climate types are significant ($p < 0.01$) for both soil moisture and hydrological droughts. Further analysis based on Tukey's honestly significant difference tests shows that pairwise differences in mean rank SPI-n of almost all climate types are statistically significant ($p < 0.01$) with just one exception. The mean rank SPI-n for winter hydrological droughts in continental and polar climates are not significantly different. Similarly, the differences in propagation time between summer and winter droughts for all drought types are significantly different based on Chi-squared and paired t-tests ($p < 0.05$), except for the paired t-test for soil moisture droughts in tropical wet climates. Despite the fact that most group means differ significantly, they are not always substantially different in magnitude. For example, the difference between mean summer and winter SPI-n for runoff droughts in temperate climates is very small.

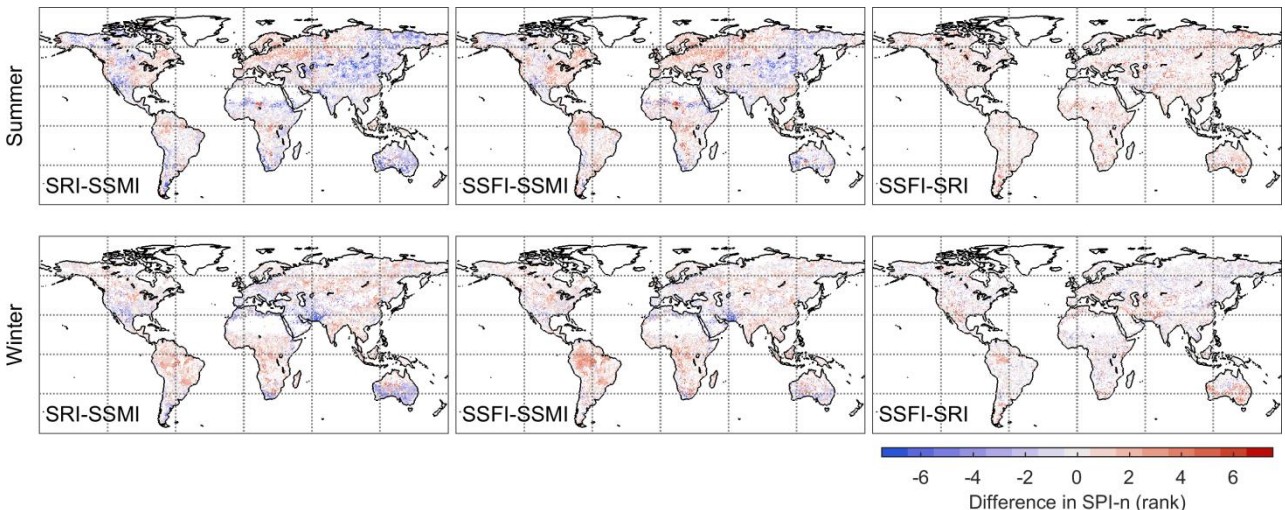

**Figure 4: The difference in the rank of SPI-n for SRI and SSMI, SSFI and SSMI, SSFI and SRI. Pixels where the difference between accumulation periods are not statistically significant (p < 0.05) are masked.**

The difference between the timescales of drought propagation to soil moisture and hydrological droughts can provide additional insights into the mechanisms of drought propagation. The differences in the rank of SPI-n are shown in Fig. 4. As explained in Sect. 2.2, we use rank SPI-n rather than the duration in months because these are more useful in interpreting differences between (sub-)seasonal and annual timescales than the SPI-n in months. A difference of 1–2 rank SPI-n indicates that drought propagation occurs at similar timescales (i.e. sub-seasonal, seasonal or yearly time scales). Differences of more than four rank SPI-n represent large differences in drought propagation timescales, such as between sub-seasonal and yearly timescales. In summer, SPI-n for runoff are higher than for soil moisture in the Amazon, eastern North America, central Africa, and parts of Europe. This means that drought propagation to soil moisture is quicker than drought propagation to runoff, which suggests that subsurface runoff or baseflow is more important than surface runoff in these locations. The opposite is true for most parts of Australia, large parts of central and eastern Asia, and parts of western North America. In

these locations, drought propagation to soil moisture drought is slower than drought propagation to runoff, which implies that surface runoff is an important component of total runoff. The differences can be substantial, with rank differences of five and more, which roughly represent the difference between sub-seasonal and annual timescales. In winter, differences tend to be smaller, and longer drought propagation timescales for runoff than soil moisture (i.e. positive values in Fig. 4) are more common. Drought propagation timescales for streamflow droughts tend to be longer than for runoff drought, which is consistent with streamflow being routed runoff.

Spearman correlation coefficients show that the difference in rank SPI-n of summer soil moisture and runoff droughts is negatively related to the amount of surface runoff relative to total runoff ($\rho = -0.53$). The relationships with average annual precipitation ($\rho = 0.44$) and the runoff coefficient ($\rho = 0.36$) are positive but slightly weaker. Each of the correlation coefficients noted here is highly significant ($p < 0.01$). The amount of surface runoff relative to total runoff, annual average precipitation, and the runoff coefficient are also related to the difference between soil moisture and streamflow drought, though the relationships are slightly weaker (correlations are up to 0.1 lower). The difference in SPI-n between drought types in winter, as well as between runoff and streamflow droughts in summer, cannot be explained by these variables.

Results of propagation analyses from soil moisture to runoff and streamflow drought, and from runoff to streamflow drought are largely consistent with the results shown in Fig. 4. For example, regions where meteorological drought propagates into soil moisture drought at shorter time scales than it does to runoff or streamflow drought (i.e. red colors in Fig. 4) are represented by longer timescales of SSMI-n (Fig. S3). Regions showing negative values in Fig. 4, on the other hand, tend to be best represented by SSMI-n of one month, or the shortest accumulation period. As discussed previously, hydrological drought may not be preceded by soil moisture drought in these regions: the negative values suggest that surface flow may be more important than subsurface flow, thereby bypassing the soil moisture store. Therefore one-month SSMI-n appear to be the most appropriate, even though the mechanism of drought propagation through surface flow cannot be represented when analyzing the propagation from soil moisture to hydrological drought. Propagation from runoff to streamflow drought largely occurs at short time scales of one to two months, which is consistent with the similarity of the global patterns of SPI-n for these drought types in Figs. 3 and 4.

In an additional analysis, we calculated SPI-n for the model ensemble mean focusing on mild droughts (SI $\leq -0.5$) and moderate droughts (SI $\leq -1$). When using mild droughts as a threshold value, the results are largely similar to those including near-normal conditions (SI $\leq 0$), as shown in Fig. S4. Both the global patterns as well as the differences between the climate types are similar to the results shown in this section. However, the number of pixels masked due to insignificant correlations between SPI-n and the SSMI, SRI or SSFI increases. When moderate droughts are used as a threshold (Fig. S5), the proportion of pixels that are masked increases further, resulting in 40–50 % less data than when including near-normal conditions. In addition, the maps of SPI-n become noisier and the distributions of SPI-n by climate type (as shown in Fig. 3) flatten, especially in continental and polar climates. However, the relationships between the climate and seasonal group means remain similar.

Additional sensitivity tests were based on the number of SPI accumulation periods studied and the method used to calculate the ensemble mean. Global patterns of SPI-n (as shown in Figure 2) and the difference in rank SPI-n (as shown in Figure 4) are very similar when fewer SPI accumulation periods are used (1, 3, 6, 12 and 24 months). The outcomes of the Chi-squared and ANOVA tests investigating the differences in SPI-n by climate type are also statistically significant and

therefore unchanged. Similarly, calculating the ensemble mean based on the original model time series rather than SI time series has a limited effect on the global patterns of soil moisture and on the average rank SPI-n per climate type. A higher occurrence of longer SPI-n for soil moisture droughts was associated with a substantially higher average soil moisture content and soil moisture variability in one or two models. The model ensemble mean based on averaging soil moisture time series was therefore more sensitive to soil moisture conditions in those one or two models than in the other models (Fig. S1).

The patterns of runoff drought propagation found in this study are similar to a previous study that calculated correlations between ensemble median runoff and precipitation percentiles (van Huijgevoort et al., 2013). Specifically, runoff is more closely related to shorter SPI (or precipitation percentile) accumulation periods in tropical regions, and to longer accumulation periods in continental and polar climates. However, more specific regional comparisons between these studies are hindered by differences in the approach of the two studies. In Van Huijgevoort et al. (2013), correlations are based on the

full precipitation and runoff time series, while in our study they are limited to below-average runoff conditions in either summer or winter months only. The results shown here are also in line with results from an observational study comparing SPI and SSFI in the United Kingdom performed by Barker et al. (2016). Barker et al. (2016) reported that SPI-n of 1–4 months were most closely related to SSFI, except in the southeast where some major aquifers are located and where longer SPI-n were found. That is just slightly shorter than the 2–6 months found in this study. Where drought propagation occurs at

very short timescales, drought is mainly driven by precipitation deficits, possibly in combination with temperature anomalies (though these are not reflected in the SPI). Where drought propagation occurs at long timescales, attenuation by hydrological stores likely plays a more important role.

## 4.2 Model variability

The variability of SPI-n in the models underlying the model ensemble mean is shown in Fig. 5, while individual model

results can be found in Fig. S6 (summer) and Fig. S7 (winter). Again, the standard deviation is not shown in months, but in the rank of SPI-n timescales studied. In summer, the standard deviation ranges from 1–4 rank SPI-n. Note that differences of 1–2 rank SPI-n indicate that models tend to agree on whether drought propagation occurs at sub-seasonal, seasonal, or annual timescales. Model variability is low in temperate regions such as Europe and eastern North America, as well as in tropical regions such as the Amazon and Southeast Asia. The model variability is high in (semi-)arid regions such as the

Sahel and central Australia. The patterns of model variability in hydrological drought propagation timescales are largely similar to those of soil moisture drought. However, the variability is patchier, and there are some regional differences. For example, model variability in SPI-n in tropical and temperate climates is slightly higher for runoff than for soil moisture. In addition, model variability is lower in central Asia for runoff droughts than for soil moisture droughts.

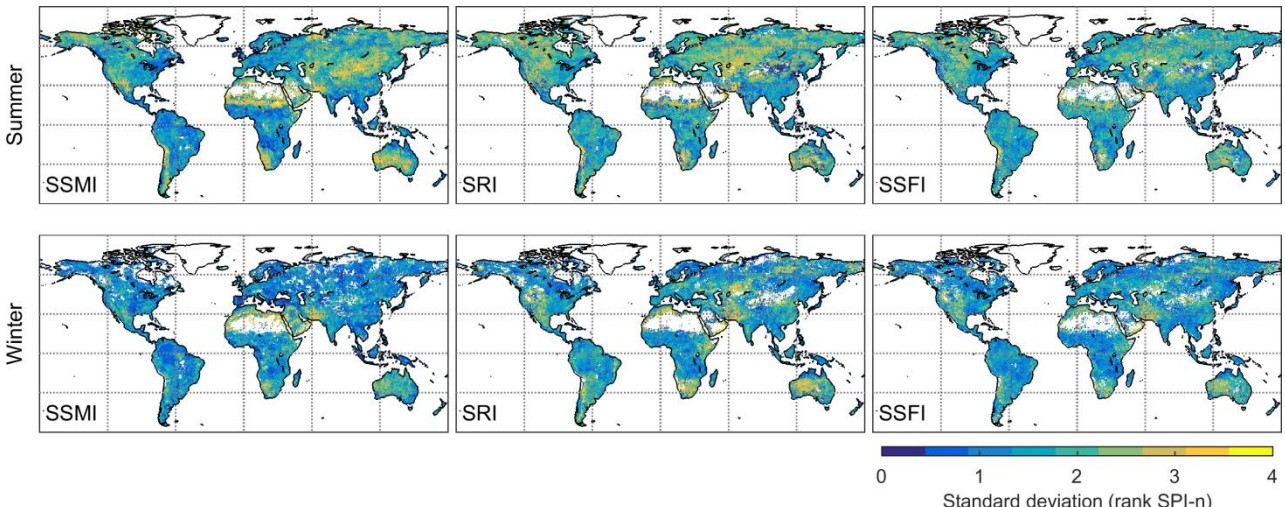

**Figure 5: The standard deviation of the rank of SPI-n between the different hydrological models for summer and winter droughts in SSMI, SRI, and SSFI.**

Attributing observed differences between models to model characteristics is not straightforward, despite the common meteorological forcing, because there are considerable differences in model structures and parameterizations (Beck et al., 2016, 2017; Döll et al., 2016). However, we examine the relationship between drought propagation and specific model characteristics to identify areas for further study. First, we examine the relationship between SPI-n and average soil moisture storage, as previous work has suggested that water storage plays an important role in drought propagation (Barker et al., 2016; Van Loon and Laaha, 2015). In addition, we examine the effect of model structural choices on drought propagation in an exploratory analysis.

Soil moisture storage in the models varies considerably between models, mainly due to differences in the definitions of root-zone soil moisture used to calculate the SSMI. The reported depths of the root zone range from 0.2 to 8 m, which may be a fixed value for all pixels or vary by pixel and/or land use type. Although we use standardized indices (SSMI) and not absolute values of soil moisture, the response time to changes in precipitation and/or evaporation will differ between soils with large and small storage volumes. We examine the relationship between average root-zone soil moisture storage and average SPI-n for soil moisture droughts per climate type in Fig. 6. Soil moisture storage is averaged over space and time, and SPI-n in space, resulting in a single point for each model. The figure shows that drought propagation from meteorological to soil moisture drought is strongly related to average soil moisture storage, with correlation coefficients between 0.56 and 0.91, depending on the season and climate type. The impact of changes in storage on drought propagation is especially high in dry climates, where relatively small differences in average soil moisture correspond to large differences in SPI-n. In comparison, the impact of storage on SPI-n is low in tropical wet climates. In general, changes in SPI-n with storage are smaller in winter than in summer. For tropical wet, continental, and polar climates the impact of storage is less

than one rank SPI-n over the full range of storage volumes in winter, suggesting that storage is not an important driver of model variability in this season.

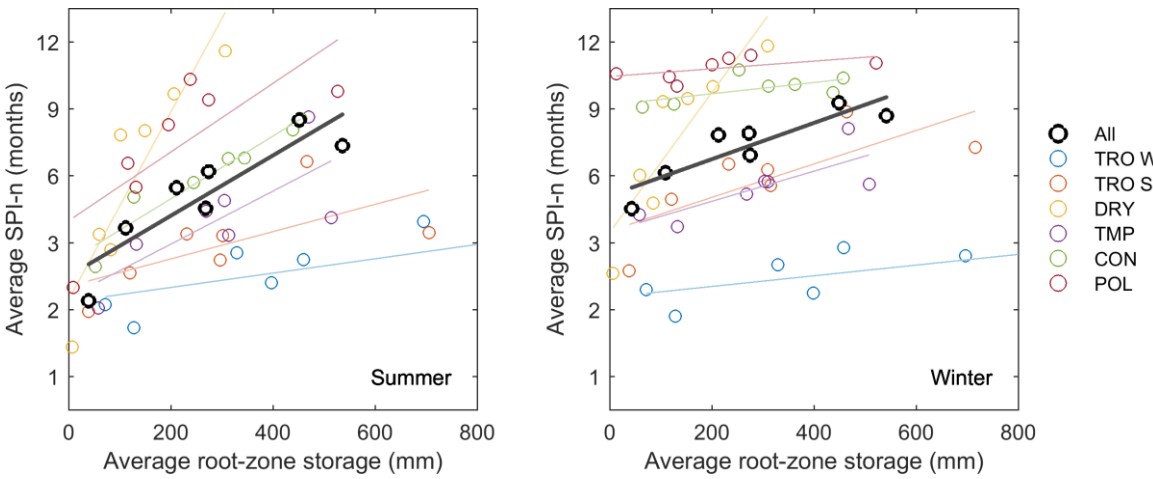

 **Figure 6: SPI-n for the SSMI averaged over all pixels and each climate type separately plotted against mean annual root-zone soil moisture storage averaged over the same region. Each point represents a model. Lines of best fit have been added for reference.**

The relationship between soil moisture storage and drought propagation to runoff and streamflow is not included here because the link is not as clear-cut. Runoff consists of surface and subsurface components. The subsurface component of runoff is related to water stored in groundwater and/or in the lowest layer of the soil, and not to the upper soil layers included in the root zone. Furthermore, groundwater data are not available for three out of seven models. The surface component, on the other hand, is only affected by soil moisture in as much that saturated conditions near the surface will result in a larger surface flow component. Therefore, it is impacted by the relative saturation of the soil rather than the storage itself.

In an exploratory analysis, we also examine the relationship between four qualitative factors related to model structure and the timescales of drought propagation. First, we compare SPI-n in LSMs and GHMs. Second, we study the effect of different evaporation schemes, specifically comparing Penman-Monteith evaporation schemes to more empirical temperature-based approaches. Third, we group models by runoff generation mechanisms, comparing models that include infiltration excess runoff generation to those that only represent saturation excess. In this last analysis, we exclude ORCHIDEE and WaterGAP3 because they use alternative methods of runoff generation; a Green-Ampt infiltration and a beta function, respectively. Note that the limited number of models and large number of different model parameterizations and structural choices means that we cannot definitively attribute observed differences between models to any of these characteristics. Instead, this analysis is meant to be an initial exploration of the results.

The relative importance of these model characteristics based on Cohen's d (see Sect. 2.2) varies considerably over the different climates and drought types (Fig. 7). Overall, the tested model characteristics are associated with larger effects on soil moisture droughts than on runoff and streamflow droughts. Grouping models by model type has the largest effect on

mean soil moisture drought SPI-n for most climates, where drought propagation is slower in LSMs than in GHMs. For runoff droughts, on the other hand, GHMs tend to have higher SPI-n for runoff droughts than LSMs. However, we cannot determine which underlying or associated model characteristics are the primary sources of the difference between the groups in this analysis.

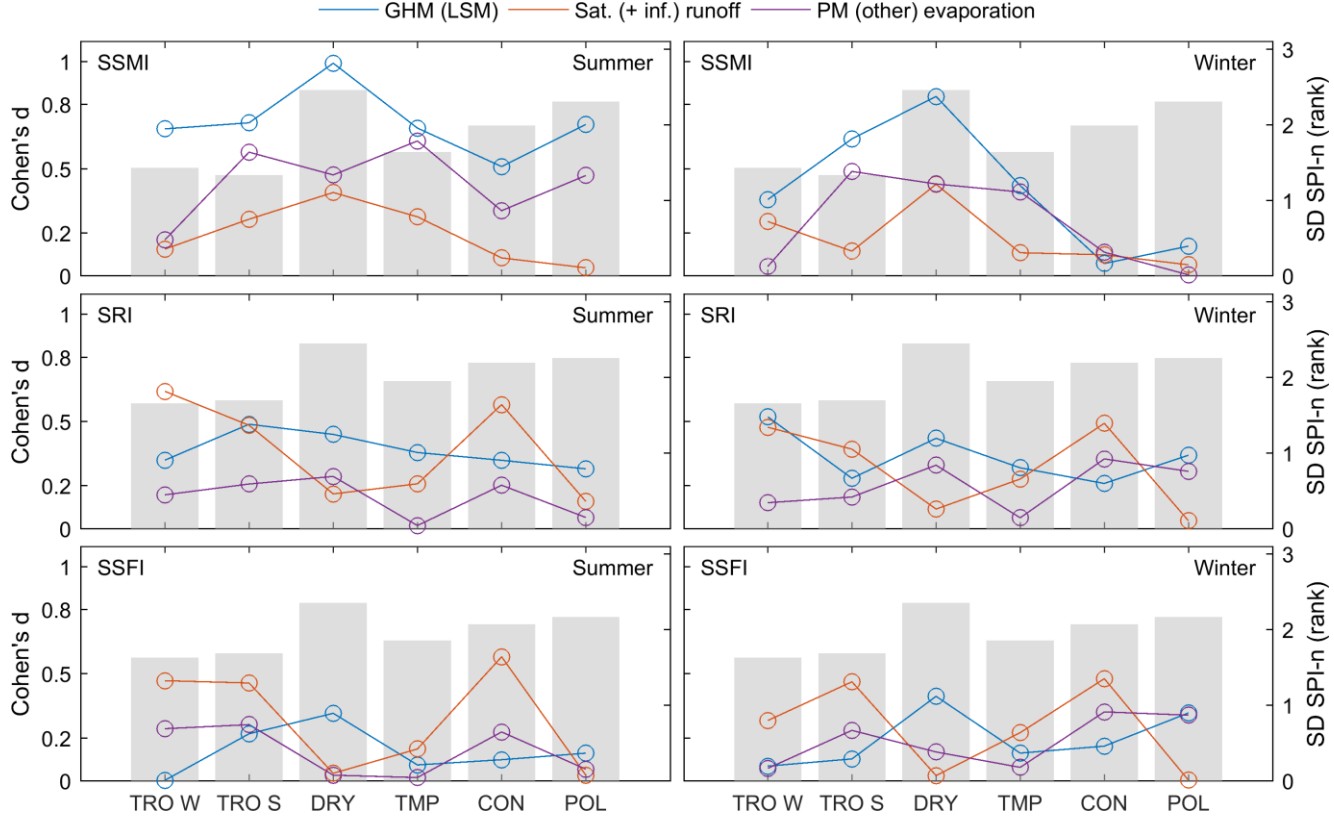

**Figure 7: Effect sizes based on Cohen's d for model structural choices by climate for summer and winter droughts in SSMI, SRI, and SSFI. Bars represent the standard deviation of model SPI-n for each group. Abbreviations of climate types are the same as in Fig. 1.**

Grouping models by runoff generation mechanisms has a smaller effect on average SPI-n for soil moisture droughts than the other studied factors. However, it can be more relevant for runoff and streamflow droughts, especially in tropical and continental climates. In these climates, including infiltration excess runoff leads to lower SPI-n and faster drought propagation. In tropical climates, this can be explained by the fact that high-intensity rainfall events exceeding the infiltration capacity of the soil are more common.

Note that the model characteristics tested here are not fully independent. The number of models is small compared to the number of differences in model structures and parameterizations. Therefore, grouping models by different characteristics can result in identical groups, making it impossible to untangle the two in the current study setup. One example is the snow scheme. Previous studies have suggested that using energy-based or temperature-based snow schemes results in different

model behavior, both based on the same models used in this study (Beck et al., 2017) as well as in other models (Haddeland et al., 2011). In this study, however, it is impossible to distinguish between model type and snow scheme since all LSMs use an energy-based approach, while GHMs use temperature-based approaches. Another example is simulation of reservoirs. Previous studies have recognized that human adaptation should be taken into account in drought studies (Van Loon et al., 2016; Veldkamp et al., 2015) and that the simulation of reservoirs has a substantial impact on drought propagation (Lorenzo-Lacruz et al., 2013). However, grouping models by whether they simulate reservoirs results in nearly identical groups as dividing them by model type. In fact, only W3RA is classified as another group.

Insight into which factors are truly responsible for model differences can only be gained through exhaustive experiments testing different model parameterizations and structures (i.e. Medlyn et al., 2015). This type of analysis was unfortunately not possible within the experimental setup of this study because model parameterizations were not consistent between models (see Sect. 3). Nevertheless, while the effect sizes cannot be used to confirm that a certain model characteristic is the true factor underlying observed differences, they can be used to identify directions for further study in more comprehensive analyses.

## 4.3 Evaluation against observations

The timescales of drought propagation from meteorological to streamflow drought in the models and model ensemble mean have been evaluated against data from 127 in-situ streamflow stations (Fig. 8). In this way, we can investigate whether the modeled drought propagation times presented in Sect. 4.1 and 4.2 resemble reality. Of the individual models, W3RA performs best for summer droughts and WaterGAP3 for winter droughts based on MAE and Spearman correlations between modeled and observed SPI-n. The performance of the model ensemble mean is similar to that of the best-performing model in both seasons, with the highest correlations and (nearly) the lowest mean absolute errors with observed SPI-n. WaterGAP3 is the only model that was calibrated against streamflow observations, which could be a reason for the good performance in winter, even though it is outperformed by all other models in summer. While the mean absolute errors of the ensemble mean and individual models are within one or two rank SPI-n, correlations are low for all models.

In addition to the overall performance metrics, we compare the number of models for which the difference with observed SPI-n is small (absolute error $\leq 1$) and the number of models for which this difference is large (absolute error $\geq 4$) at each site (Fig. 8). The thresholds are based on the models' (in)ability to capture the overall timescales of drought in terms of sub-seasonal, seasonal or yearly timescales. At least six out of seven models are within one step of the observed SPI-n at 16 and 19 % of the study sites in summer and winter, respectively, and at least four models are within one step at 45 and 39 % of the study sites for summer and winter, respectively. This suggests that models are well able to capture observed drought propagation timescales at these sites. However, the majority of models differ by at least 4 steps of SPI-n at approximately 10 % of sites. At these sites, the models show substantially different drought propagation timescales. These differences in SPI-n correspond to differences between sub-seasonal and annual timescales of drought propagation. Models tend to do well in western North America and Europe, but poorly in central North America and parts of South America and Australia.

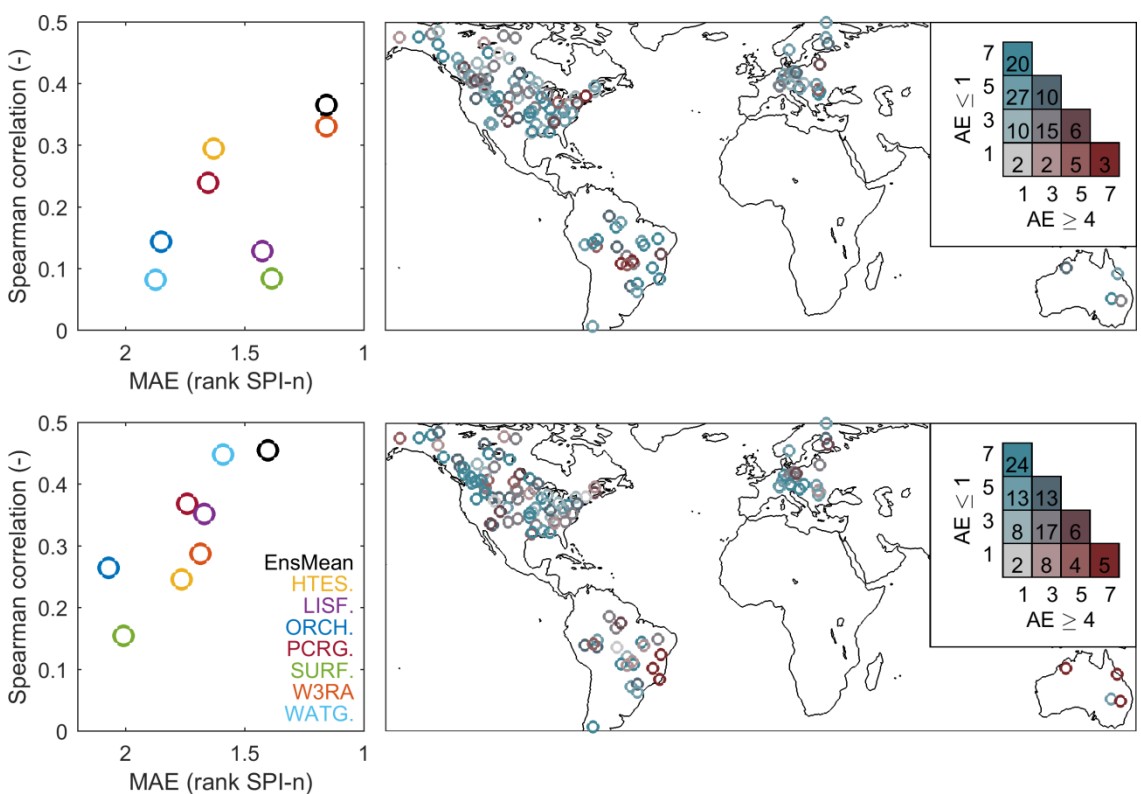

**Figure 8: Comparison of model and observed summer (top) and winter (bottom) streamflow drought propagation timescales. The Spearman correlation and mean absolute error based on rank are compared, where better performing models fall in the upper right corner. A map shows the number of models where the absolute error is ≤ 1 or ≥ 4 (right). The number of represented sites is indicated in the symbols of the legend.**

We use the mean error (ME) rather than the MAE to investigate the relationship between errors in rank SPI-n and climate type (Fig. 9). While MAE is a more suitable metric for evaluating the overall performance of the models since it is indifferent to the direction of the errors, ME allows us to assess whether models are more likely to over- or underestimate SPI-n and is thus reflects model bias. On average, most models and the model ensemble mean tend to overestimate summertime SPI-n in tropical wet climates, which is also the climate for which mean errors are largest. Models tend to underestimate SPI-n in tropical savanna climates, while results are more mixed in the other climate types (Fig. 9). The confidence intervals of the mean are smallest in continental and temperate climates, which is most likely because a relatively large number of sites are located in these climates compared to the tropical and dry climate types. However, the observed differences in mean rank SPI-n error between climate types are not always statistically significant based on Chi-squared and ANOVA tests ($p < 0.05$). Mean errors in summertime rank SPI-n are significantly different between climate types in only one model (PCR-GLOBWB) according to Chi-squared tests, and in two (HTESSEL-CaMa and PCR-GLOBWB) out of seven models, as well as the model ensemble mean, according to ANOVA tests. In winter, Chi-squared tests show that the

differences in mean error rank SPI-n are statistically significant for all but one model (LISFLOOD) as well as the ensemble mean. The results of ANOVA tests are significant for four models (HTESSEL-CaMa, ORCHIDEE, SURFEX-TRIP and WaterGAP3), but not for the model ensemble mean. Further analysis based on Spearman correlations showed no relationship between errors in rank SPI-n and catchment size.

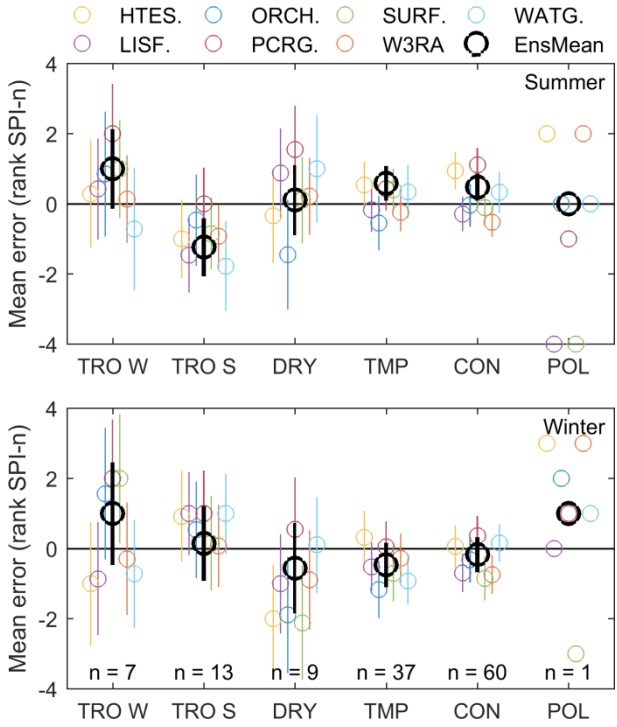

**Figure 9: Mean error in rank SPI-n between model and observed summer (top) and winter (bottom) streamflow droughts. Error bars indicate the 95 % confidence intervals of the mean, and the number of sites within each climate type group is indicated in the bottom panel. Abbreviations of climate types are the same as in Fig. 1.**

It is important to note that some of the streamflow time series span as little as 18 years, which is shorter than the 30 years of data recommended for the calculation of SI. This means that the observational time series at some sites are too short to capture the climatology of their locations. However, the average time series length (29 years) is close to the required 30 years. Furthermore, even where time series are relatively short we can evaluate whether the models capture the relationship between observed SPI and SSFI during the available time period. Another limitation of the evaluation is that the sites with observed data are not spread evenly across the globe, as most sites are located in the United States or Europe, with scarce data in Africa and Asia. Differences between the models and observations can be attributed to several types of errors (Van Loon, 2015). The first type of error concerns errors in the model meteorological forcing data, but also in the GPCC precipitation data used to create the validation dataset. Then there are the errors in model structure and parameterizations of hydrologic processes, including the representation of anthropogenic influence on streamflow. Finally, there are errors in the (discretization of) the routing schemes employed by the models.

Unfortunately, evaluation of soil moisture drought propagation timescales is inhibited by the lack of root-zone soil moisture data at global scale. While satellite soil moisture products are available, these are limited to the upper few centimeters of the soil (Owe et al., 2008), which is not representative of root-zone soil moisture. Terrestrial water storage data from the Gravity Recovery and Climate Change mission has also been used to investigate drought propagation (Zhao et al., 2017). However,

the root-zone soil moisture signal cannot be untangled from the other terrestrial water stores and therefore cannot be used for validation in this study. Field-measured soil moisture is also available, for example through the International Soil Moisture Network (Dorigo et al., 2011). However, only a handful of sites remain after applying the same site selection procedure as for streamflow drought validation (i.e. sufficiently long time series and a statistically significant relationship between SPI and SSMI).

**5 Conclusion**

This study evaluates timescales of drought propagation from meteorological to soil moisture and hydrological drought in an ensemble of seven land surface and global hydrological models. Drought propagation was quantified by cross-correlating standardized indices of hydrological variables. Here, we focus on soil moisture, runoff and streamflow droughts in summer and winter. However, the simple and flexible approach used here can be applied to other drought types, such as groundwater

droughts, and to other months or seasons.

Drought propagation is closely related to climate type, with slower drought propagation in dry and continental climates and quicker drought propagation in tropical climates. Winter season drought propagation tends to be slower than in the summer, especially in tropical savanna and continental climates. This may be a result of the distinct wet and dry seasons in the former, and snow cover in the latter. Faster propagation of meteorological drought to runoff drought than to soil moisture drought

has been linked to a higher proportion of surface runoff, thereby causing a larger portion of total runoff to bypass the soil moisture store.

Model variability can be quite high, especially for summer droughts and in dry climates, where the socio-economic impacts of drought can be severe. Since the models were run with consistent forcing datasets, differences can be attributed to model parameterization and structure. For example, drought propagation from meteorological to soil moisture drought was

generally slower in models with higher average soil moisture storage, and vice versa. Although the differences cannot be definitively attributed to specific model characteristics in the current experiment, we identified several directions for further study. Grouping models by model type and runoff generation mechanisms is especially promising, as these factors are associated with significantly different average drought propagation timescales. A true physical interpretation of the results would require comprehensive experiments in which model structural choices and parameterizations are consistently changed

for all participating models. However, this was not possible in the experimental set-up.

The relationship between meteorological and streamflow drought in the global models was evaluated against observational data. Overall, the models were able to capture the timescales of drought propagation, as errors were relatively low on

average. However, considerable model advancements can be made since there were large discrepancies between model and observed drought propagation at 10 % of the study sites.

A better understanding and representation of drought propagation in global models may improve drought forecasting (Cancelliere et al., 2007), especially when combined with the availability of accurate seasonal forecasts (Luo et al., 2007; Luo and Wood, 2007). Drought forecasting potential is expected to be higher in regions with relatively slow drought propagation, such as the dry and continental climates in this study, as drought forecasting for longer SPI-n tends to be more accurate than for shorter SPI-n (Mishra and Desai, 2005). Additional research using lagged SPI-n could assess the potential for forecasting different types of drought based on meteorological data. Improved representation of drought propagation in models is also crucial to constrain the impact of climate change on drought frequency and severity, and thereby improve the reliability of projected changes.

**Data availability**

The meteorological forcing, model outputs, and standardized indices from the eartH2Observe project are openly available from the project portal (www.earth2observe.eu). GPCC daily precipitation data can be obtained via https://www.esrl.noaa.gov/psd/data/gridded/data.gpcc.html and GRDC monthly streamflow data are available at http://grdc.bafg.de.

**Acknowledgements**

This research received funding from the European Union Seventh Framework Programme (FP7/2007-2013) under grant agreement no. 603608, Global Earth Observation for integrated water resource assessment: eartH2Observe.

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
