# Peer review of "The effect of climate type on timescales of drought propagation in an ensemble of global hydrological models"

_Hydrology and Earth System Sciences, 2017_

## Referee Comment (RC1) · Anonymous Referee #1 · 15 Jan 2018

Review of "The effect of climate on timescales of drought propagation in an ensemble of global hydrological models" by Gevaert et al.

In this study a model ensemble is used to characterize the propagation timescales from meteorological drought to soil moisture drought and to hydrological drought. The propagation times are analysed in respect of climate, season and model type and evaluated with observed streamflow. This is an interesting, well-structured study, with some substantial conclusions that will be a useful addition to literature. The supplements, tables and figures (apart from Figure 3, see comments below) are self-explaining and appropriate. However, I have three main issues with the methodological approach, that

should be addressed before publication.

Main issues: The authors chose for their analyses eight accumulation periods that represent different timescales (1, 2, 3, 6, 9, 12, 24 and 36 months for sub-seasonal, seasonal and annual timescales). These accumulation periods are similar to those that are often used, but still arbitrary. For example, they could have chosen only 1, 3, 6, 12 and 24 months for the same reasons. For the determination of propagation timescales this choice is probably of minor relevance, however, for the applied statistical tests I think it can have quite some impact. The tests used in this study are designed for variables on the interval scale (apart from spearman's rho) but the variables are on ordinal scale. The authors are aware of that problem and state they "assume that the difference between accumulation periods of 12 and 24 months (...) to be equivalent to the difference between 1 and 2 months" (p.5, l.5ff). Nevertheless, it is still very relevant to check whether there is an influence of the arbitrary choice of accumulation periods and the related assumption on the results of the statistical tests. Additionally, it needs a strong rationale for using tests designed for interval scaled variables instead of tests appropriate for ordinal scaled variables (e.g. the chi-squared test of independence).

The second main issue is about the way the model ensemble mean is calculated: "The model ensemble mean was calculated as the average of the SIs" (p.6, l.29). A very important reason for using standardised indices is to ensure that all time series have the same distribution and are directly comparable (see e.g. Bloomfield and Marchant, 2013; Kumar et al., 2016). Averaging two or more timeseries, that have a standard normal distribution, will lead to a timeseries which distribution has a smaller standard deviation that might favour certain (higher) SPI-n. Moreover, the comparison with the results from the original model time series as it is carried out in chapter 4.3 is not really "fair" anymore, since time series are not directly comparable. The correct way is to average the raw model outputs first and standardize afterwards all seven time series plus the model ensemble mean.

Finally, the authors use an explanatory analysis to identify relevant model character-

istics causing differences in drought propagation timescale (p.15f). They are aware of the difficulties using only seven models for that and the problem of collinearity between the groups. In fact, these limitations inhibit any useful result. For example, the factors GHM/LSM and (no)reservoir are highly correlated. Based on Table 1, it is only the model W3RA that is classified into another group. That means, in a study without this model, the groups would have been identical, similar to what is reported about the snow scheme. Accordingly, the graphs in Figure 7 of the two groups have a very similar shape. The authors wonder about the reason for the high influence of reservoirs on soil moisture (p.16, l.12), but the real problem is, that both factors (GHM/LSM and (no)reservoirs) represent the combined effect of (no) reservoirs, GHM/LSM, snow scheme and probably several other relevant model structures. As it is not possible to relate the differences of the groups to one model structure we cannot learn much from this analysis.

Other points the authors might want to look at: In the introduction the authors acknowledge that an important component of drought propagation is the time lag (p.2, l.13ff). However, time lags are not considered in the analysis but listed to be important for future research (p.19, l.16). Including an analysis of time lags which also might differ for the models would increase the relevance of this study. If lags are not included, there should be at least a rationale for excluding them despite their relevance.

In chapter 4.1 analyses of the "mean SPI-n" are presented (e.g. p.10, l.17; caption of Figure 3). For me it does not become clear, whether this is really the arithmetic mean of the SPI-n or rather the mean of the ranks. For example in Figure 3: If there were the two accumulation periods of 1 and 36 months, is "mean SPI-n" (1+36)/2=18.5 or rather (1+6)/2=3.5? This is quite relevant for the plotted circles. If they are calculated as an arithmetic mean, it might be very hard to read the values from the plot due to the very non-linear y-axis.

Moreover, it is important to report somewhere the 'sample size', i.e. the absolute number of cells which are not masked for the different climates and drought types. Otherwise it is for example hard to understand, that the t-test leads to significant different SPI-n means for winter and summer in runoff of TMP (Figure 3).

On page 10, l.16 the authors describe the results of the ANOVA: "The means of SPI-n for winter hydrological droughts in continental and polar climates are not significantly different". Again, for me it is not clear whether the rank mean or the arithmetic mean is meant here. However, more important is the fact that it sounds like two categories were directly compared to each other. In this case, it would have been a t-test rather than an ANOVA what was used. Please clarify, which variables were used for the ANOVA and in which cases a t-test was used.

The stations used for the evaluation against observations are distributed very uneven (as the authors write on p.18, l.7). In Figure 8 it looks like there were very few to no stations in the climates polar and tropical wet. However, the authors state that "errors between models and observations are not related to climate". To enable the reader to comprehend this important finding, I think it is necessary to give more information on the number of stations per climate zone, the test used to reach this conclusion as well as the results of the test.

References: Bloomfield, J. P., & Marchant, B. P. (2013). Analysis of groundwater drought building on the standardised precipitation index approach. Hydrology and Earth System Sciences, 17, 4769-4787. Kumar, R., Musuuza, J. L., Van Loon, A. F., Teuling, A. J., Barthel, R., Ten Broek, J., Mai, J., Samaniego, L., and Attinger, S. (2016). Multiscale evaluation of the Standardized Precipitation Index as a groundwater drought indicator Hydrology and Earth System Sciences, 20, 1117-1131.

---

## Short Comment (SC1) · 16 Jan 2018

This is a very interesting study on the timescale of drought propagation and is pertinent to my research interest. Here I have two minor points:

1) In the last paragraph of section 4.3, the authors state that "Unfortunately, evaluation of soil moisture drought propagation timescales is inhibited by the lack of root-zone soil moisture data at global scale. While satellite soil moisture products are available, these are limited to the upper few centimeters of the soil (Owe et al., 2008), which is not representative of root-zone soil moisture." However, satellite observations of total water storage (TWS) from the Gravity Recovery and Climate Change (GRACE) include soil

moisture variations in the root-zone. It also has been used to investigate drought propagation timescale in combination with SPEI and other satellite records at the global scale (Zhao et al. 2017 https://doi.org/10.1175/JHM-D-16-0182.1). Therefore, the authors should phrase their statement more carefully here.

2) Does the magnitude and duration of a specific meteorological drought affect the timescale of drought propagation in the terrestrial hydrologic cycle? For example, in a region a meteorological drought might happen very quickly and intense (such as flash drought), it might quickly deplete soil moisture and streamflow therefore may better agree with short time-scale SPI. But in the same region, a meteorological drought can form slowly and gradually propagate into the terrestrial hydrological cycle therefore may better agree with long time-scale SPI. Can the authors explain how this would affect the interpolation of a" temporally averaged" SPI-n presented in the paper?

---

## Referee Comment (RC2) · C. Prudhomme (Referee) · 10 Mar 2018

**General**

The paper deals with the very interesting topic of drought propagation through the water cycle, but viewed only from a climatic perspective, with analyses between precipitation and each of three land surface responses (soil moisture, runoff, and routed discharge). The analysis is reported by climatic regions and winter/summer seasons, with the aim to find commonalities in the precipitation-land surface response. The bulk of the work is done based on ensemble mean indices from a range of global hydrological/land surface models, with at the end an attempt to look at the variability in the responses by

individual models.

The subject is very topical and relevant for publication in HESS. However, I regret that the analysis is done : 1) following climatic lenses (precipitation vs land surface; no analysis of propagation between the different land surface responses; summary/ discussion based on climatic regions without attempt to relate to soil/land surface/ bedrock/ catchment size etc... components). This is a shame and a more comprehensive analysis would be more valuable. Note that the title suggest 'effect of climate' but only precipitation (and not temperature/ evaporative losses) are considered, so it is not a full climate analysis that is undertaken ; 2) primarily on a multimodel mean (smoothing out extremely different behaviour; making extremely difficult a physical interpretation of results); 3) without justification of the choice of accumulations periods, which are arbitrary.

I don't feel the manuscript can be published in its current form, but owing to the importance of the subject for the scientific community, I believe it has potential for publications if the following points are addressed appropriately:

1. Undertake a full propagation analysis, by adding correlation between land surface components (soil moisture and runoff; soil moisture and discharge; runoff and discharge), and provide physically-based/ model structure/ parameterisation interpretation of the results. The analysis should also include at minimum catchment size, and if possible information on the land surface fields that should be available for all models.

2. Change the emphasis of the paper to individual models results, with the multi-model mean analysis presented last (if at all) with a justification of what it tells us. I am curious to know how different are the average SIs compared with individual models, and what mean SI represents physically. Understanding how the structure of the models influence drought propagation would be extremely valuable for future analysis. I fully agree with the point made by Referee #1 that there are strong collinearity between the different categories used to divide the models, and this should be considered in the

interpretation of the results

3. Better justification of the choice of accumulation periods, which are very arbitrary: how different would be the results if different / additional accumulation periods were used? Ideally, a sensitivity analysis should be conducted. Are the statistical metrics used appropriate? (Point also raised by Referee #1) Whilst I understand the rationale, I struggle very much with the analysis of the 'difference in ranks' as they are really arbitrary. For example I very much like fig 3 but find fig 4 might be greatly dependent on the arbitrary accumulation periods.

4. I find difficult to understand the rationale and use of the evaluation section, as there are no real links with the rest of the analysis/ discussion/ interpretation. I think it is great to have it, but it should be more prominent. Moreover, as the authors mention, the analysis is extremely skewed with a very unequal distribution of catchments geographically. A filtering, with much fewer catchments in US and western Europe should be done. The drainage area of the model extracted points should also be compared with the catchment one. How do the stations relate to the climate zones?

5. The method section needs to be re-written, especially the section on timescale propagation, and the rationale and description of the difference analysis p5 l9 to 20; what does mean 'statistical significance test does not reflect the relevance of differences between groups'? What is the group mean (mean correlation? something else?) in equation 1 and 2? The section on evaluation of drought propagation also needs clarifying. Are the RMSE done on daily or monthly streamflow? How well the drainage area of the pixel matches that of real catchment? What model results have to be recalculated and why?

---

## Author Comment (AC1) · 26 May 2018

We thank the referee for his/her in-depth review and constructive comments. The referee raises some important points which have helped us improve the manuscript. We address each of the comments in a point-by-point fashion and have provided a revised version of the manuscript in the supplement.

Please also note the supplement to this comment: https://www.hydrol-earth-syst-sci-discuss.net/hess-2017-745/hess-2017-745-AC1-supplement.zip

---

## Author Comment (AC2) · 26 May 2018

Thank you for the effort taken to share your insightful points about our manuscript. We address each of the comments in a point-by-point fashion in the supplement. A copy of the revised manuscript is also included in the supplement.

Please also note the supplement to this comment: https://www.hydrol-earth-syst-sci-discuss.net/hess-2017-745/hess-2017-745-AC2-supplement.zip

745, 2018.

---

## Author Comment (AC3) · 26 May 2018

We would like to thank referee C. Prudhomme for the time and effort spent to review our manuscript and for her thorough and valuable comments. The feedback has helped us improve our manuscript. In the supplement, we respond to each of the comments in a point-by-point fashion and indicate what changes have been made to the manuscript. A revised version of the manuscript has also been included in the supplement.

Please also note the supplement to this comment:

[Figure]

https://www.hydrol-earth-syst-sci-discuss.net/hess-2017-745/hess-2017-745-AC3-supplement.zip

---

## Author Response (AR1)

**Response to Referee 1**

We thank the referee for his/her in-depth review and constructive comments. The referee raises some important points (in bold), and we address each of these in a point-by-point fashion below. See the second Supplement for a revised version of the manuscript.

**Main issues:**

**The authors chose for their analyses eight accumulation periods that represent different timescales (1, 2, 3, 6, 9, 12, 24 and 36 months for sub-seasonal, seasonal and annual timescales). These accumulation periods are similar to those that are often used, but still arbitrary. For example, they could have chosen only 1, 3, 6, 12 and 24 months for the same reasons. For the determination of propagation timescales this choice is probably of minor relevance, however, for the applied statistical tests I think it can have quite some impact. The tests used in this study are designed for variables on the interval scale (apart from spearman's rho) but the variables are on ordinal scale. The authors are aware of that problem and state they "assume that the difference between accumulation periods of 12 and 24 months (. . .) to be equivalent to the difference between 1 and 2 months" (p.5, l.5ff). Nevertheless, it is still very relevant to check whether there is an influence of the arbitrary choice of accumulation periods and the related assumption on the results of the statistical tests. Additionally, it needs a strong rationale for using tests designed for interval scaled variables instead of tests appropriate for ordinal scaled variables (e.g. the chi-squared test of independence).**

First, we will discuss the choice for certain statistical tests, then we will investigate the sensitivity of the results and conclusions to the (number of) accumulation periods.

As the referee indicated, there are statistical tests that have been designed for ordinal variables, such as Chi-squared and Cramer's V (effect size metric). In the preparation of this study we considered using these metrics and calculated their results. The outcome of Chi-squared, for example, was that difference in SPI-n by climate type is highly significant (p < 0.001) for all drought types and seasons. However, an important disadvantage of using Chi-squared and other metrics for ordinal data is that these metrics treat ordinal variables as categorical variables. This means that the relationships between SPI-n are ignored. In the end we chose ANOVA tests because we believe it is important to take the relationship between SPI-n into account and because the SPI accumulation periods are nearly equidistant in log space. In the revised version of the manuscript, we include outcomes of the Chi-squared metric (P12 L1+6) and elaborate on the motivation for using ANOVA tests (P5 L17-21).

The sensitivity of the conclusions to the SPI accumulation periods is a good point. To address this, we recalculated the (significance of) the results using fewer SPI accumulation periods (1, 3, 6, 12 and 24 months, as suggested). As expected, changes to global patterns of SPI-n are minimal when fewer accumulation periods are used. This is shown for summer SPI-n in Figure R1, but is also true for winter SPI-n. In addition, outcomes of Chi-squared and ANOVA tests are still highly significant (p < 0.001). The pairwise comparisons using Tukey's honestly significant difference test show minor differences for runoff droughts. To be more specific, the difference in mean rank SPI-n between

tropical savanna and dry climates is no longer statistically significant in summer. In winter, the difference between tropical wet and dry climates is no longer statistically significant. Pairwise t-tests were used to test between summer and winter droughts, and the results for fewer accumulation periods were the same as with more accumulation periods. Therefore, the conclusions of this study are not affected by using fewer SPI accumulation periods. We summarize the results of this sensitivity test in the revised version of the manuscript (P14 L2-5).

[Figure]

**Figure R1.** The SPI accumulation period (SPI-n) resulting in the highest correlations with model ensemble mean SSMI, SRI, and SSFI, for summer droughts using the original larger selection of accumulation periods (top) and a smaller selection of accumulation periods (bottom). Pixels where those correlations are not statistically significant ($p < 0.05$) are masked.

**The second main issue is about the way the model ensemble mean is calculated: "The model ensemble mean was calculated as the average of the SIs" (p.6, l.29). A very important reason for using standardised indices is to ensure that all time series have the same distribution and are directly comparable (see e.g. Bloomfield and Marchant, 2013; Kumar et al., 2016). Averaging two or more timeseries, that have a standard normal distribution, will lead to a timeseries which distribution has a smaller standard deviation that might favour certain (higher) SPI-n. Moreover, the comparison with the results from the original model time series as it is carried out in chapter 4.3 is not really "fair" anymore, since time series are not directly comparable. The correct way is to average the raw model outputs first and standardize afterwards all seven time series plus the model ensemble mean.**

We agree that we have deviated from the usual way of calculating the ensemble mean by averaging SI time series rather than the original model time series. Our motivation for averaging SI time series was that we did not want one or two models with high soil moisture/discharge (variability) to dominate the overall signal. For discharge we expected this to be less of an issue than soil moisture, where total storage (and variability) varies considerably between models (see for example Figure 6 of the manuscript). In addition, though standardization is an important reason for using SIs, we do not use the time series directly in the analyses. Even in the evaluation section, we compare SPI-n and not the time series themselves.

Nevertheless, we recalculated ensemble mean SPI-n using the approach of averaging of the original model time series. Overall, results are similar to when the ensemble mean was based on averaging SI time series: SPI-n does not change in 60-67% of all pixels, and changes by a maximum of 1 rank SPI-n in 80-85% of all pixels (Figure R2). For all climates and seasons, the mean rank SPI-n changes by less than 1. A closer examination of the pixels showing a change in SPI-n shows that soil moisture droughts are most affected by changing the ensemble mean calculation method. For these droughts, SPI-n tends to be higher when original model time series rather than SI time series are averaged. This is especially the case for summer droughts in tropical wet, dry and polar climates.

[Figure]

 **Figure R2.** Histograms of the change in rank SPI-n when the ensemble mean is calculated as the average of the original model time series compared to when the SI time series are averaged. Changes in rank SPI-n are shown by climate type, and for summer and winter droughts in SSMI, SRI, and SSFI. Circles represent the mean change in rank SPI-n per climate type and season.

To further investigate the somewhat higher SPI-n for soil moisture droughts, we studied a pixel with a tropical wet climate located in central Africa (Figure R3). ORCHIDEE and SURFEX-TRIP, the models with the highest average soil moisture conditions (largely due to a deeper definition of the root zone), have a much higher soil moisture variability than the other models. The SI time series of these models are also very different from those of the other models. For example, the drought between 2004 and 2007 is much more pronounced in ORCHIDEE and SURFEX-TRIP, and the time series are smoother. As a result of the higher soil moisture variability, SURFEX-TRIP and ORCHIDEE have a larger impact on the average of the original model time series (EnsMean 2), and thus also in the resulting SI time series. Averaging SI time series (EnsMean 1) is a better representation of the 'average' behavior within the model ensemble.

The results shown in Figure R3 are representative of other model pixels where changing the ensemble mean calculation method results in changes of more than 3 rank SPI-n. The underlying cause of these large differences is that the models use different definitions for root-zone soil moisture. In some models this is a fixed depth, in others this varies with vegetation type. Ideally, the root-zone soil moisture time series could be normalized between 0 and the maximum soil moisture content before further analyses. However, the maximum soil moisture content is not always easy to define because vegetation types and rooting depths can vary within pixels.

[Figure]

**Figure R3.** Time series of soil moisture content relative to the multi-year mean (top) and SSMI (bottom) for each of the individual models and two methods of calculating the ensemble mean. EnsMean 1 is based on averaging model SI time series, EnsMean 2 is based on averaging the original model time series.

In summary, though changing the calculation of the ensemble mean can have a large impact on SPI-n for individual pixels, the main conclusions of this study are not sensitive to the ensemble mean calculation method. This is probably because even though averaging does result in fewer extreme values in the ensemble mean, SPI-n are based on correlations between SI time series, which are not as sensitive as other metrics to a narrower range in values. Since the results are similar overall and due to the results of the soil moisture averaging analysis as shown in Figure R3, we have decided to still calculate the model ensemble based on SI time series. However, we have added a statement clarifying why we chose a non-standard method to calculate the ensemble mean (P7 L22-29 and Fig. S1 in the revised manuscript), and that the main conclusions of this study are not impacted by this choice (P14 L5-9).

**Finally, the authors use an explanatory analysis to identify relevant model characteristics causing differences in drought propagation timescale (p.15f). They are aware of the difficulties using only seven models for that and the problem of collinearity between the groups. In fact, these limitations inhibit any useful result. For example, the factors GHM/LSM and (no)reservoir are highly correlated. Based on Table 1, it is only the model W3RA that is classified into another group. That means, in a study without this model, the groups would have been identical, similar to what is reported about the**

**snow scheme. Accordingly, the graphs in Figure 7 of the two groups have a very similar shape. The authors wonder about the reason for the high influence of reservoirs on soil moisture (p.16, l.12), but the real problem is, that both factors (GHM/LSM and (no)reservoirs) represent the combined effect of (no) reservoirs, GHM/LSM, snow scheme and probably several other relevant model structures. As it is not possible to relate the differences of the groups to one model structure we cannot learn much from this analysis.**

We completely agree that we cannot attribute the observed differences to investigated model structures and parameterizations. We attempted to make it clear that while we cannot do so, we present an initial exploration only. For example, we think it is important to note the large differences between LSMs and GHMs, even though we cannot pinpoint the exact mechanism(s) responsible for that difference. This exploratory nature of this analysis has been made clearer in the results section (P16 L19-21) and in the conclusion (P21 L25-30).

Our paragraph concerning the simulation of reservoirs was poorly phrased. We included this factor because previous studies have shown that it plays a role in hydrological drought propagation. We agree that reservoirs are not likely to play a role in soil moisture droughts, and therefore by including it we intended to warn that apparent differences can be misleading. In the revised manuscript, we have removed this factor from the effect size figure (Figure 7 in the manuscript). In the text, we instead refer to previous studies that investigated reservoirs and hydrological drought and explain that this distinction is not useful in our case due to the high similarity with grouping by model type (P18 L3-7).

**Other points the authors might want to look at: In the introduction the authors acknowledge that an important component of drought propagation is the time lag (p.2, l.13ff). However, time lags are not considered in the analysis but listed to be important for future research (p.19, l.16). Including an analysis of time lags which also might differ for the models would increase the relevance of this study. If lags are not included, there should be at least a rationale for excluding them despite their relevance.**

Lag is indeed an important characteristic of drought propagation. We have recalculated SPI-n using SPI with different accumulation periods as well as lags up to 12 months for the model ensemble mean. Results show that, as expected, taking lag into account has a larger impact on winter drought propagation than on summer drought propagation. In summer, the best drought propagation result has a lag of 0 months in around 70 % of the pixels (Figure R4). In winter, 0-month lags are still the most common, but account for only about 40 % of the pixels. Shorter lags (1–3 months) are more prevalent than longer lags (8–12 months).

The frequent occurrence of 0-month lags for summer drought propagation means that SPI-n also remains unchanged in a majority of pixels. However, even for winter droughts SPI-n are not affected by taking lags into account in about 60 % of the pixels. This means that for about 20 % of pixels, taking lag into account does not change SPI-n, but does improve the correlation between SPI-n and the winter drought SI of interest. Changes in SPI-n in both seasons tend to be small, with changes in rank SPI-n larger than 2 occurring in less than 10 % of pixels. Positive changes in SPI-n are more

frequent than negative changes, meaning that overall taking lag into account leads to longer SPI-n. This is the opposite of what we hypothesized in the previous version of the manuscript. We had expected that lag would be especially important in areas with significant snow cover in winter, and that including lags would lead to lower SPI-n.

These results show that lag does play a role in drought propagation, though this is more so in winter than in summer. Even so, SPI-n are not very sensitive to whether lags are included in the analysis or not. We could add this figure to the supplementary material, but the Supplement is already large, containing seven figures. Since this was not one of the major comments, we suggest not adding this figure to the supplement. However, we leave the final decision to the referee and editor.

[Figure]

**Figure R4.** The SPI-n lag in months leading to the best correspondence with SSMI, SRI and SSFI for summer and winter droughts (top) and the change in rank SPI-n compared to when lags are not taken into account (bottom). Pixels where the correlations between lagged SPI-n and SI time series are not statistically significant (p < 0.05) have been masked.

**In chapter 4.1 analyses of the "mean SPI-n" are presented (e.g. p.10, l.17; caption of Figure 3). For me it does not become clear, whether this is really the arithmetic mean of the SPI-n or rather the mean of**

the ranks. For example in Figure 3: If there were the two accumulation periods of 1 and 36 months, is "mean SPI-n" (1+36)/2=18.5 or rather (1+6)/2=3.5? This is quite relevant for the plotted circles. If they are calculated as an arithmetic mean, it might be very hard to read the values from the plot due to the very non-linear y-axis.

This should indeed all be mean rank SPI-n. We have changed "mean SPI-n" to "mean rank SPI-n".

Moreover, it is important to report somewhere the 'sample size', i.e. the absolute number of cells which are not masked for the different climates and drought types. Otherwise it is for example hard to understand, that the t-test leads to significant different SPI-n means for winter and summer in runoff of TMP (Figure 3).

Agreed, we have added the number of pixels for each climate and drought type to the panels in Figure 3 of the revised manuscript.

On page 10, l.16 the authors describe the results of the ANOVA: "The means of SPI-n for winter hydrological droughts in continental and polar climates are not significantly different". Again, for me it is not clear whether the rank mean or the arithmetic mean is meant here. However, more important is the fact that it sounds like two categories were directly compared to each other. In this case, it would have been a t-test rather than an ANOVA what was used. Please clarify, which variables were used for the ANOVA and in which cases a t-test was used.

This should indeed be mean rank SPI-n, and the text has been revised to reflect this. An ANOVA test was used for the comparison over multiple groups. A statistically significant ANOVA test result was followed by Tukey's honestly significant difference tests to compare each pair of group means. This test is very similar to a t-test, but corrects for family-wise error rates. This correction is needed because the chance of making a type 1 error (false positive) increases when comparing multiple groups. The use of Tukey's tests has been added to the Methods section (P5 L21-23) and the Results section (P12 L2-4).

The stations used for the evaluation against observations are distributed very uneven (as the authors write on p.18, l.7). In Figure 8 it looks like there were very few to no stations in the climates polar and tropical wet. However, the authors state that "errors between models and observations are not related to climate". To enable the reader to comprehend this important finding, I think it is necessary to give more information on the number of stations per climate zone, the test used to reach this conclusion as well as the results of the test.

We agree that the link between GRDC stations and climate type is not clear. Therefore, we have included the stations in Figure 1 of the manuscript (the global map of Köppen Geiger classification used). In addition, we have included the number of stations falling within each class in the legend of the same figure.

The relationship between error in SPI-n and climate zone was based on ANOVA tests (p < 0.05). In the previous version of the manuscript, ANOVA results were not significant for any of the models. In

the revised version of the manuscript, however, some of the results have changed because we included an additional criteria for GRDC stations based on the agreement in upstream catchment area (see referee 2, comment 4). ANOVA results are now statistically significant for two out of seven models in summer and four models in winter. For the model ensemble mean, a statistically significant result is only found for summer droughts. The explanation of the test used and its results has been added to this section (P19 L7 – P20 L4, Fig. 9).

**Response to referee C. Prudhomme**

We would like to thank referee C. Prudhomme for the time and effort spent to review our manuscript and for her thorough and valuable comments. The feedback has helped us improve our manuscript. Below, we respond to each of the comments and indicate what changes have been made to the manuscript. The revised version of the manuscript has been included as a supplement.

**The subject is very topical and relevant for publication in HESS. However, I regret that the analysis is done :**

**1) following climatic lenses (precipitation vs land surface; no analysis of propagation between the different land surface responses; summary/ discussion based on climatic regions without attempt to relate to soil/land surface/ bedrock/ catchment size etc. . . components). This is a shame and a more comprehensive analysis would be more valuable. Note that the title suggest 'effect of climate' but only precipitation (and not temperature/ evaporative losses) are considered, so it is not a full climate analysis that is undertaken ; 2) primarily on a multimodel mean (smoothing out extremely different behaviour; making extremely difficult a physical interpretation of results); 3) without justification of the choice of accumulations periods, which are arbitrary.**

**1. Undertake a full propagation analysis, by adding correlation between land surface components (soil moisture and runoff; soil moisture and discharge; runoff and discharge), and provide physically-based/ model structure/ parameterisation interpretation of the results. The analysis should also include at minimum catchment size, and if possible information on the land surface fields that should be available for all models.**

> We have performed the suggested full propagation analysis by investigating drought propagation from soil moisture to both types of hydrological drought, and from runoff to streamflow drought. Similar to the analysis of propagation from meteorological to streamflow drought, the soil moisture and runoff to streamflow drought analyses used catchment-aggregated values of soil moisture and runoff. In summer, SSMI-1 had the best agreement with SRI and SSFI for most of world (Figure R5). Slightly higher SSMI-n are found in tropical regions, northern Europe, and parts of North America. In winter, longer SSMI-n up to (multi-)annual scales are more common, especially in continental climates. Drought propagation from runoff to streamflow droughts is generally quick in both seasons, with over 90 % of the pixels having a SRI-n less than or equal to 2 months. The similarity between runoff and streamflow drought propagation time scales is consistent with the meteorological drought propagation results (Figure 3 of the manuscript) and the difference in SPI-n (Figure 4 of the manuscript) for these drought types. Furthermore, the regions with positive differences between SPI-n in Figure 4 of the manuscript correspond more or less to the regions where SI-n are longer than 1 month in Figure R1 below, though the magnitudes are not necessarily equal. On the other hand, negative values in Figure 4 of the manuscript usually correspond to SI-n of one month. In these regions, drought propagation to runoff or streamflow is quicker than to soil moisture, which may be linked to a larger surface flow component in total runoff. This results in one-month SPI-n, though in fact this type of drought mechanism largely bypassing soil moisture

cannot be captured when analyzing propagation of soil moisture drought to hydrological drought. The full drought propagation analysis is presented in Figure S3 of the revised manuscript and described in an additional paragraph in Section 4.1 (P13 L14-24).

[Figure]

**Figure R5.** The SSMI and SRI accumulation period (SSMI-n or SRI-n) resulting in the highest correlations with model ensemble mean SRI and SSFI, for summer and winter droughts. Pixels where those correlations are not statistically significant (p < 0.05) are masked.

We fully agree that a physically based interpretation of differences between models would be very insightful, but is unfortunately not possible within eartH2Observe. Such an interpretation would require extensive experiments changing a large number of model structures and parameterizations, for example using Monte Carlo analyses. Even then, prescribing different sets of parameterizations is further complicated because choice of a certain parameterization is closely linked to the modeling system. This is also the reason the project did not prescribe a fixed set of static fields. We emphasize the need for comprehensive model structure and parameterization experiments in the Results (P16 L1-3) and Conclusion (P21 L28-30) sections of the revised manuscript.

The "effect of climate" in the title was meant to reflect how many of the analyses in our study focus on differences in drought propagation between Köppen-Geiger climate types. To make this clearer, we have changed the title to "The effect of climate **type** on timescales of drought propagation in an ensemble of global hydrological models".

**2. Change the emphasis of the paper to individual models results, with the multi-model mean analysis presented last (if at all) with a justification of what it tells us. I am curious to know how different are the average SIs compared with individual models, and what mean SI represents physically. Understanding how the structure of the models influence drought propagation would be extremely valuable for future analysis. I fully agree with the point made by Referee #1 that there are strong collinearity between the different categories used to divide the models, and this should be considered in the interpretation of the results.**

The ensemble mean result gives us an idea of the model consensus on drought propagation time scales globally and how these differ per climate type. We expect that the individual model results are indeed of interest to the respective modeling groups because they can compare their results to other models and the model ensemble mean, or model consensus. Therefore, we have added the model-specific results to the Supplementary Material (Figures S6 and S7). However, to make individual model results insightful for the larger community we would need to be able to attribute the observed differences to model structures and/or parameterizations. As explained previously in response to the first comment, this is not possible within the current experimental setup.

We agree that we cannot use the categories we used to divide the models to definitively identify the mechanisms underlying differences in SPI-n due to the large number of potential factors and limited number of models (or the collinearity between groups). Instead, we attempted an initial exploration of potential explanations for the differences between models based on our observations and previous work. We have rephrased the introduction to this analysis (P16 L19-21) to better reflect this, and also modified the way we refer to this analysis in the conclusion section (P21 L25-28). In addition, we have removed the (no) reservoir groups from Figure 7 in the revised manuscript (Cohen's d effect size) due to the similarity with the LSM/GHM groups. We use the (no) reservoir group as another example of a factor that was found to be important in previous studies, but which we cannot isolate in our study (P18 L3-7).

**3. Better justification of the choice of accumulation periods, which are very arbitrary: how different would be the results if different / additional accumulation periods were used? Ideally, a sensitivity analysis should be conducted. Are the statistical metrics used appropriate? (Point also raised by Referee #1) Whilst I understand the rationale, I struggle very much with the analysis of the 'difference in ranks' as they are really arbitrary. For example I very much like fig 3 but find fig 4 might be greatly dependent on the arbitrary accumulation periods.**

We apologize that we did not make our choice of SPI accumulation periods clearer. The accumulation periods were based on those commonly used, where possible adding intermediate values to allow more subtle differences to be observed. These accumulation periods are furthermore nearly equidistant in log space.

Additional analyses were performed to determine whether using fewer SPI accumulation periods (1, 3, 6, 12 and 24 months) would impact the main conclusions of this study. As shown in Figure R6, the global patterns of SPI-n are not greatly impacted by this choice. In addition, the global patterns of the difference in rank SPI-n (Figure 4 of the manuscript) are very similar when fewer accumulation periods are used (Figure R7). The values and range of the difference in rank SPI-n change due to the smaller number of accumulation periods, but overall the direction and relative magnitude are similar. In this way, reducing the number of accumulation periods does not have a large effect on the conclusions of this study. We have added a summary of the results of the sensitivity analysis to the revised manuscript (P14 L2-5).

Finally, we address the point related to the statistical tests. As the referee indicated, there are statistical tests that have been designed for ordinal variables, such as Chi-squared and Cramer's V (an effect size metric). However, these metrics treat ordinal variables as categorical variables, which means that the relationships between SPI-n are ignored. In the preparation of this study, we did calculate Chi-squared and found highly significant results (p < 0.001) for all tests analyzing SPI-n for different drought types by climate type or season. In the end we chose ANOVA tests because ignoring the relationship between SPI-n seemed unrealistic. This explanation has been added to the methods section (P5 L17-21). In addition, we report outcomes of Chi-squared tests (P12 L1 + 7). See also our response to referee 1's first comment.

[Figure]

**Figure R6.** The SPI accumulation period (SPI-n) resulting in the highest correlations with model ensemble mean SSMI, SRI, and SSFI, for summer droughts using the original larger selection of accumulation periods (top) and a smaller selection of accumulation periods (bottom). Pixels where those correlations are not statistically significant (p < 0.05) are masked.

[Figure]

**Figure R7.** The difference in the rank of SPI-n for SRI and SSMI, SSFI and SSMI, SSFI and SRI for the original SPI accumulation periods studied (top, same as Fig. 4 of the manuscript) and for a smaller selection of accumulation periods (bottom). Pixels where the difference between accumulation periods are not statistically significant (p < 0.05) are masked.

**4. I find difficult to understand the rationale and use of the evaluation section, as there are no real links with the rest of the analysis/ discussion/ interpretation. I think it is great to have it, but it should be more prominent. Moreover, as the authors mention, the analysis is extremely skewed with a very unequal distribution of catchments geographically. A filtering, with much fewer catchments in US and western Europe should be done. The drainage area of the model extracted points should also be compared with the catchment one. How do the stations relate to the climate zones?**

This section evaluates whether drought propagation from meteorological to streamflow drought in the models is similar to observations. Therefore, this is a first reality check for the results shown in the previous sections. We have added the rationale for the evaluation section (P3 L7-8) and added a link to the results in Sections 4.1 and 4.2 (P18 L16-17). The number of GRDC sites does vary considerably over the study sites. While this is unfortunate, removing stations to ensure there are

an equal number of stations per climate type would result in not using more than half of the observational data available.

We did not compare the model and GRDC upstream catchment areas in the first version of the manuscript. In the current version, we applied an additional criteria specifying that the model upstream catchment area may not deviate more than 25%  from the size of the GRDC catchment area. This has resulted in significantly fewer stations (126 instead of 297). Not surprisingly, the mean absolute error and Spearman correlations between modeled and GRDC rank SPI-n tend to improve (see revised version of Figure 8). The additional criteria based on the GRDC catchment  area has been added to the methods section (P8 L10-11) and the results in section 4.3 have been updated to reflect the new selection of GRDC stations.

**5. The method section needs to be re-written, especially the section on timescale propagation, and the rationale and description of the difference analysis p5 l9 to 20; what does mean 'statistical significance test does not reflect the relevance of differences between groups'? What is the group mean (mean correlation? something else?) in equation 1 and 2?**

We try to distinguish between "statistical significance" and "relevance" of the difference between groups. That is to say that with a large number of observations as we have in this study, even very small differences between group means can be statistically significant. This sentence has been rephrased in the revised manuscript (P5 L25-27).

The group mean in equation 1 and 2 is the mean rank SPI-n for a specific climate type. This is now specified in the revised manuscript (P6 L1).

**The section on evaluation of drought propagation also needs clarifying. Are the RMSE done on daily or monthly streamflow? How well the drainage area of the pixel matches that of real catchment? What model results have to be recalculated and why?**

We agree that this section could use some clarification. The RMSE based on monthly streamflow data is used to assign a GRDC station to a model pixel. The streamflow data have been evaluated in previous work (Beck et al., 2017; Schellekens et al., 2016), therefore our evaluation focuses only on the drought propagation, or SPI-n. As stated in response to the previous comment, we have added an additional criteria for GRDC site selection to ensure that the model catchment area is within 25% of the GRDC catchment area. This information has been added to the Methods and Data sections.

The model SPI-n have to be recalculated in the evaluation section because there can be missing values in the observational time series. To ensure we compare like with like, we recalculate the model SI time series, and resulting SPI-n, using only months in which observational data are available. We have rephrased the sentence to make this clearer (P6 L19-21).

References

[revised manuscript text omitted]